



# The Relationship between Lower Stratospheric Ozone in the

# Southern High Latitude and Sea Surface Temperature in the

# East Asia Marginal Seas

Wenshou Tian[1], Yuanpu Li[1], Fei Xie[2*], Jiankai Zhang[1], Martyn P. Chipperfield[3],

Wuhu Feng[4], Sen Zhao[5], Xin Zhou[6], Yun Yang[2], Xuan Ma[2]

[1]*College of Atmospheric Sciences, Lanzhou University, Lanzhou, China*

[2]*College of Global Change and Earth System Science, Beijing Normal University, Beijing, China*

[3]*ICAS, School of Earth and Environment, University of Leeds, Leeds, UK*

[4]*NCAS, School of Earth and Environment, University of Leeds, Leeds, UK*

[5]*Key Laboratory of Meteorological Disaster of Ministry of Education, and College of Atmospheric*

*Science, Nanjing University of Information Science and Technology, Nanjing, China*

[6]*State Key Laboratory of Numerical Modeling for Atmospheric Sciences and Geophysical Fluid*

*Dynamics, Institute of Atmospheric Physics, Chinese Academy of Sciences, Beijing, China*

Submitted as an Article to: ***Atmospheric Chemistry and Physics***

*Corresponding author:

Dr. Fei Xie, Email: xiefei@bnu.edu.cn.





**Abstract**
Using satellite observations, reanalysis data, and model simulations, this study
investigates the effect of sea surface temperatures (SST) on interannual variations of
lower stratospheric ozone in the southern high latitude. It is found that the SST
variations across the East Asian marginal seas (5 °S–35 °N, 100 °E–140 °E) rather than
the tropical eastern Pacific Ocean, where ENSO occurs, have the most significant
correlation with the southern high latitude lower stratospheric ozone changes. Further
analysis reveals that planetary waves originating over the marginal seas can be
propagated to southern middle to high latitudes via two teleconnection pathways in
summer and one pathway in autumn. The anomalous propagation and dissipation of
ultra-long Rossby waves in the stratosphere strengthen/cool (weaken/warm) the
southern polar vortex which produces more (less) active chlorine and enhances
(suppresses) ozone depletion in the southern high latitude stratosphere on one hand,
and impedes (favors) the transport of ozone from the southern middle latitude
stratosphere to high latitude on the other. The model simulations also reveal that
approximately 17% of the decreasing trend in the southern high latitude lower
stratospheric ozone observed over the past five decades can be attributed to the
increasing trend in SST over the East Asian marginal seas.





## 1. Introduction

Ozone variations over recent decades exhibit not only a strong decreasing trend, forced by changes in ozone-depleting substances superimposed on a changing climate, but also interannual variability influenced by various external and internal climate forcings (e.g. Manney et al. 1994; Müller et al., 1994, 2005; Weiss et al., 2001; Hadjinicolaou et al., 2002; Tian and Chipperfield, 2005; Austin et al., 2006, 2010; Eyring et al., 2010; Liu et al., 2011, 2013; Douglass et al., 2014). Ozone variations can change the amount of harmful solar ultraviolet rays reaching the Earth's surface (Kerr and McElroy, 1993) and even influence climate (Forster and Shine, 1997; Thompson et al., 2011; Li et al., 2016; Xie et al., 2016). Therefore, clarifying the processes that are responsible for ozone variability is crucial for understanding how global climate interacts with ozone variations (Austin et al., 2006; Hess and Lamarque, 2007; Frossard et al., 2013; Rieder et al., 2013). Many previous studies have analyzed the ozone variability caused by external processes such as volcanic aerosols (e.g. Hofmann and Oltmans, 1993; Rozanov et al., 2002; Dhomse et al., 2015) and the solar cycle (e.g. Chandra and McPeters, 1994; Rozanov et al., 2005; Dhomse et al., 2016) and these studies showed that volcanic aerosols and solar variations can result in considerable short- and long-term variations in ozone levels. Ozone variations can also be caused by changes in the surface climate (Zhang et al., 2014). Other studies have reported the effects of internal climate variability on ozone, including El Niño–Southern Oscillation (ENSO; Ziemke and Chandra, 1999; Cagnazzo et al., 2009; Randel et al., 2009; Xie et al., 2014a, 2014b; Zhang et al., 2015a, 2015b), Madden–Julian Oscillation (MJO; Fujiwara et al., 1998; Tian et al., 2007; Liu et al., 2009; Weare, 2010; Li et al., 2012), Arctic Oscillation (AO) or North Atlantic Oscillation (NAO; Schnadt and Dameris, 2003; Lamarque and Hess, 2004; Creilson et al., 2005;





Steinbrecht et al., 2011), and Quasi-Biennial Oscillation (QBO; Bowman, 1989; Tung and Yang, 1994; Dhomse, 2006; Li and Tung, 2014). These studies indicate that ozone over different regions shows different variability due to the location-specific nature of the processes that influence this variability.

The stratospheric ozone hole (Farman et al., 1985) over the Antarctic has been shown to have an important impact on the Southern Hemisphere climate (Shindell and Schmidt, 2004; Son et al., 2008, 2009, 2010, Perlwitz et al., 2008; Feldstein, 2011, Kang et al., 2011, Polvani et al., 2011; Thompson et al., 2011; Cagnazzo et al., 2013; Keeble et al., 2014; Previdi and Polvani, 2014). Although the principal mechanisms responsible for the formation of the ozone hole are well understood (e.g., Solomon, 1990, 1999; Ravishankara et al., 1994, 2009), the factors or processes that generate interannual variations in ozone levels in the southern high latitude stratosphere remain under debate. Among various factors, the QBO has been reported to have a significant impact on the interannual variations of the Antarctic ozone (Garcia and Solomon, 1987; Lait et al., 1989; Mancini et al., 1991; Gray and Ruth, 1993; Bodeker and Scourfield, 1995; Shindell et al., 1997a). The September to March levels of ozone over the Antarctic is also marginally correlated with the wintertime mean eddy heat flux (Weber et al., 2003). Heat transport induced by upward propagating planetary waves warms the polar vortex (Schoeberl and Hartmann, 1991), which reduces the occurrence of polar stratospheric clouds (PSCs), a key prerequisite for the heterogeneous chemistry that depletes Antarctic ozone. Subsequent efforts to understand Antarctic ozone variations during individual years have considered planetary wave activity, which account for much of the interannual variations of ozone levels over the Northern Hemisphere (Hadjinicolaou et al., 1997; Fusco and Salby, 1999; Salby and Callaghan, 2004, 2007a, 2007b; Hadjinicolaou and Pyle,





2004). Studies based on measurements (Bodeker and Scourfield, 1995), modeling
(Shindell et al., 1997a, 1997b), and reanalysis data (Huck et al., 2005) have shown
that interannual differences in the severity of Antarctic ozone depletion are
anti-correlated with Southern Hemisphere planetary wave activity. However, the
source of the planetary wave activity that modulates interannual variability in
southern high latitude stratospheric ozone is still not well understood.

7         Variations in tropical sea surface temperatures (SST) associated with El

Niño-Southern Oscillation (ENSO), are an important factor in the modulation of the
planetary wave activity in the Northern Hemisphere that affects the interannual
variability of temperature and ozone levels in the northern polar stratosphere (Sassi et
al., 2004; Manzini et al., 2006; Calvo et al., 2004, 2009; Cagnazzo et al., 2009; Hu
and Pan, 2009; Hurwitz et al., 2011a, b; Ren et al., 2010; Zubiaurre and Calvo, 2012;
Xie et al., 2012; Yao et al., 2015). The long-term trend in tropical SST also has
corresponding to the trend of temperature in the southern polar stratosphere (Grassi et
al., 2005, 2006; Hu and Fu, 2009; Li et al., 2010; Clem et al., 2016). Although ENSO
is reported to cause circulation and temperature anomalies in the southern high
latitude stratosphere, the interannual variability of the southern polar vortex and ozone
levels over the past three decades cannot be explained by ENSO variations alone
(Angell, 1988, 1990; Hurwitz et al., 2011a, 2011b; Lin et al., 2012; Wilson et al., 2014;
Evtushevsky et al., 2015; Yu et al., 2015; Yang et al., 2015; Welhouse et al., 2016).

21        Over recent decades, SST in the East Asian marginal seas has followed an

increasing trend with strong interannual variations (Zheng et al., 2014). Zhao et al.
(2015, 2016) pointed out that Rossby waves generated by variations in the SST of the
South China Sea can cross the equator and be propagated towards to southern middle
to high latitudes. It is likely that the Rossby waves generated by SST changes in the



vicinity of the East Asian marginal seas can cross the equator to the Southern
Hemisphere and regulate ozone levels in the southern high latitude stratosphere via
their influence on the southern stratospheric circulation. Therefore, it is worthwhile to
examine the potential connections between SST variations over the East Asian
marginal seas and southern high latitude lower stratospheric ozone variations. The
remainder of the paper is organized as follows. The data, method and model used are
introduced and briefly described in section 2. Section 3 analyzes the connection
between the East Asian marginal seas and southern high latitude lower stratospheric
ozone. Section 4 presents and discusses the simulations of the connection. Finally, the
results are summarized and conclusions drawn in section 5.

## 2. Data, Model, and Methods

The ozone data used in this study is obtained from the NASA Modern Era
Retrospective Analysis for Research and Applications (MERRA) dataset version 2
(Rienecker et al., 2011) and TOMCAT/SLIMCAT 3-D model simulations
(Chipperfield, 2006). The MERRA2 data uses 42 pressure levels from the surface up
to 0.1 hPa. The vertical resolution of MERRA2 is ~1–2 km in the UTLS and 2–4 km
in the middle and upper stratosphere. MERRA2 is assimilated by the Goddard Earth
Observing System Model, Version 5 (GEOS-5) with ozone from the Solar
Backscattered Ultra Violet (SBUV) radiometers from October 1978 to October 2004,
and thereafter from the Ozone Monitoring Instrument (OMI) and AURA Microwave
Limb Sounder (MLS) (Bosilovich et al., 2015). The MERRA2 reanalysis ozone data
compares well with satellite ozone observations (Rieder et al., 2014; Zhang et al.,
2015b) and shows a better representation of the QBO and stratospheric ozone
compared to MERRA1 (Coy et al., 2016). In the present study, the ozone data from a



3D offline chemical transport model, SLIMCAT (Feng et al., 2007, 2011), is also used.
The simulation performed in this study is driven by horizontal winds and temperatures
from meteorological analyses of the ERA-Interim data provided by European Centre
for Medium-Range Weather Forecasts (ECMWF) (Dee et al., 2011). The vertical
advection in the model is calculated from the divergence of the horizontal mass flux
(Chipperfield, 2006), and chemical tracers are advected by the conservation of
second-order moments (Prather, 1986). Figure 1 shows the ozone variations over the
region 200–50 hPa and 60–90 °S, where the variability and depletion of ozone
concentration is most pronounced in the Southern Hemisphere in the past five decades
(Austin and Wilson, 2006; Solomon 1990, 1999; Ravishankara et al., 1994, 2009),
from the two datasets. The ozone variations from MERRA2 are in good agreement
with those from SLIMCAT (Fig. 1a), and the difference between the two kinds of
ozone data is small (Fig. 1b).

14       SST data is obtained from HadISST dataset compiled by the UK Met Office

Hadley Centre for Climate Prediction and Research (Rayner et al., 2003).
Geopotential height, zonal wind, and temperature data are obtained from the ECMWF
ERA-Interim dataset.

18       We also use version 4 of the Whole Atmosphere Community Climate Model

(WACCM4) in this study since WACCM has been shown to simulate well the
stratospheric circulation, temperature and ozone variations (Garcia et al. 2007).
WACCM4 is part of the Community Earth System Model (CESM) framework
developed by the National Center for Atmospheric Research (NCAR). WACCM4 uses
a finite-volume dynamical core, with 66 vertical levels extending from the ground to
$4.5 \times 10^{-6}$ hPa (145 km geometric altitude), and a vertical resolution of 1.1–1.4 km in
the tropical tropopause layer and the lower stratosphere (below a height of 30 km).





The simulations presented in this paper are performed at a horizontal resolution of
$1.9° \times 2.5°$ and with interactive chemistry (Garcia et al., 2007). More details
regarding WACCM4 are provided in Marsh et al. (2013).

4        We calculate the statistical significance of the correlation between two

auto-correlated time series using the two-tailed Student's $t$-test and the effective
number ($N^{\text{eff}}$) of degrees of freedom (DOF; Bretherton et al. 1999). For this study, $N^{\text{eff}}$
is determined using the following approximation (Li et al. 2012):
$$\frac{1}{N^{eff}} \approx \frac{1}{N} + \frac{2}{N} \sum_{j=1}^{N} \frac{N-j}{N} \rho_{XX}(j)\rho_{YY}(j)$$
where $N$ is the sample size, and $\rho_{XX}$ and $\rho_{YY}$ are the autocorrelations of two sampled
time series, $X$ and $Y$, respectively, at time lag $j$.

11       We use the formulae given by Andrews et al. (1987) to calculate the

quasi-geostrophic 2D Eliassen–Palm (E–P) flux. The meridional ($F_y$) and vertical ($F_z$)
components of the E–P flux, and the E–P flux divergence $D_F$, are expressed as:
$$F_y = -\rho_0 a \cos\varphi \overline{u'v'}$$
$$F_z = -\rho_0 a \cos\varphi \frac{Rf}{HN^2} \overline{v'T'}$$
$$D_F = \frac{\nabla \cdot F}{\rho_0 a \cos\varphi} = \frac{\partial(F_y \cos\varphi)/a\cos\varphi\partial\varphi + \partial F_z/\partial z}{\rho_0 a \cos\varphi},$$
where $\rho_0$ is the air density, $a$ is the radius of the Earth, $R$ is the gas constant, $f$
is the Coriolis parameter, $H$ is the atmospheric scale height (7 km), $u$ and $v$ are
the zonal and meridional wind components, respectively, and $T$ is the temperature;
the overbar denotes the zonal mean, and the prime symbol denotes departures from
the zonal mean.



**3. The connection between the East Asian marginal seas and southern high latitude lower stratospheric ozone**

Figure 2a shows the correlation coefficients between SST and southern high latitude lower stratospheric ozone variations between 1979 and 2015 using ozone data from the MERRA2 dataset and SST from HadISST dataset. Ozone data from SLIMCAT simulations was further used to confirm the correlation coefficients (Fig. 2b). The regions of significant correlation are generally different for the two ozone datasets except for the East Asian marginal seas; i.e., 5 °S–35 °N, 100 °E–140 °E, where the most significant correlations between Antarctic stratospheric ozone variations and SST are seen in both datasets. Figure 2 implies an interannual connection between SST in the East Asian marginal seas and southern high latitude lower stratospheric ozone variations. Figure 2 also reveals that SST variations associated with ENSO are not the main factor controlling the interannual variability of southern high latitude lower stratospheric ozone.

Through the interannual connection in Fig. 2 possibly caused by the influence of lower latitude SST on the south high latitude stratosphere, south high latitude stratospheric ozone has also been shown to affect tropical climate (Son et al., 2008; Kang et al., 2011; Thompson et al., 2011). Thus, it is first necessary to confirm the causality of this connection. To investigate the SST variations across the marginal seas of East Asia, we first define an SST index over the region with the most significant correlations in Fig. 2, i.e., the ST_MSEA index. This index is a time series that represents SST variations across the marginal seas of East Asia (Figure 3a). It is calculated by averaging the SST variations in the region from 5 °S–35 °N at 100 °E–140 °E, and then removing the seasonal cycle and linear trend. Fig. 3b and c show the composite warm and cold SST anomalies for the events that occurred in the marginal





seas of East Asia between 1979 and 2015 (see Table 1).
The ST_MSEA index and southern high latitude lower stratospheric ozone
variations show a significant simultaneous correlation (Fig. 4). This implies that SST
variations in the marginal seas of East Asia have an impact on southern high latitude
lower stratospheric ozone, since there is a lag of several months associated with the
effect of southern high latitude lower stratospheric ozone on the tropical climate (Son
et al., 2008; Kang et al., 2011; Thompson et al., 2011).
It is well known that the SST changes in the eastern Pacific, the Indo-Pacific
warm pool, and the Atlantic can significantly influence the northern polar stratosphere
(Calvo et al., 2004, 2009; Hoerling et al., 2001, 2004; Cagnazzo et al., 2009; Hu and
Fu, 2009; Hu and Pan, 2009; Li et al., 2010; Hurwitz et al., 2011a, b; Lin et al., 2012;
Zubiaurre and Calvo, 2012; Xie et al., 2012; Li and Chen, 2014). SST variations in
some regions can excite Rossby wave trains and those waves can propagate into
northern middle and high latitude stratosphere (Gettelman et al., 2001; Sassi et al.,
2004; Manzini et al., 2006; Garc á-Herrera et al., 2006; Taguchi and Hartmann, 2006;
Garfinkel and Hartmann, 2007, 2008; Free and Seidel, 2009). The mechanism that
allows SST variations in the East Asian marginal seas to affect the southern high
latitude stratosphere is also possibly related to tropospheric wave propagation from
northern lower latitude to southern middle and high latitudes.
Figure 5 shows the ray paths of waves generated by the SST anomalies over the
region 5 °S–35 °N, 100 °E–140 °E, at 300 hPa in four seasons. The wavenumbers along
these rays are between 1 and 5. The wave ray paths represent the climate
teleconnections; i.e., the propagation of stationary waves in realistic flows. The
calculation of the wave ray paths and application of the barotropic model is described
in detail by Li et al. (2015) and Zhao et al. (2015). We found that the Rossby waves



generated by SST anomalies in the marginal seas of East Asia could indeed propagate
to the middle to high latitudes of the Southern Hemisphere in summer and autumn
(Fig. 5b and c), but not in spring and winter (Fig. 5a and d) because the Rossby waves
motivated by the low-latitude SST anomalies move mostly northwards in spring and
winter. Meanwhile, we must note that the propagating paths of those waves in
summer and autumn are different (Fig. 5b and c). In summer, the first path of rays
originates over the marginal seas of East Asia, crosses the Indian Ocean to arrive over
tropical Africa or even South America, and then reflects equatorward to the middle to
high latitudes of the Southern Hemisphere. The second path of rays originates over
the marginal seas of East Asia reflects directly into the southern Indian Ocean and
reaches the Southern Hemisphere. In autumn, the first path disappears, and only the
rays that follow the second path reach the Southern Hemisphere. In addition, the ray
stops at about 60 °S, which possibly implies an upward propagation of the wave at this
location.
The correlation coefficients between the ST_MSEA index and 300-hPa
geopotential height variations from the ERA-Interim reanalysis across the four
seasons are shown in Figure 6. The positive and negative centers of correlation
coefficients represent the teleconnection patterns. The teleconnection patterns in
summer and autumn (Fig. 6b and c) are in good agreement with the ray paths (Fig. 5b
and c). In summer, two clear wave train paths appear over the marginal seas of East
Asia with one moving westwards to South America and reflecting to the middle to
high latitudes of the Southern Hemisphere, and the second reflecting directly into the
Southern Hemisphere (Fig. 6b). In autumn, the first path is very distinct (Fig. 6c), i.e.,
the negative correlation coefficient over the Indian Ocean is small, which suggests
that most of the waves do not propagate westwards. The second path also remains





evident. These two teleconnection pathways of the wave trains in summer and autumn
(Figs. 5 and 6) are discussed in detail by Zhao et al., (2016), who refer to them as the
North Australia–Southern Hemisphere and South Africa–Southern Hemisphere
pathways, respectively. In spring and winter, the above two teleconnection patterns
don't exist (Fig. 6a and d).
Figure 7a shows the correlation coefficients between the ST_MSEA index and
stratospheric ozone variations, which indicate that warm (cold) SST anomalies over
the East Asian marginal seas are associated with a decrease (increase) in southern high
latitude lower stratospheric ozone. Bodeker and Scourfield (1995), Shindell et al.
(1997a, 1997b), and Huck et al. (2005) have shown that interannual differences in the
severity of southern high latitude lower stratospheric ozone depletion are related to
Southern Hemisphere planetary wave activity. All of the above analysis illustrates that
the SST anomalies over the marginal seas of East Asia are a possible main source of
this planetary wave activity.
Figure 7b shows that ST_MSEA is positively correlated with zonal wind around
60 ˚S, where is the boundary of the southern polar vortex in summer and autumn,
while Figs. 7c indicate that ST_MSEA is negatively correlated with temperature. The
correlations shown in Figs 3, 5, 6, and 7 can be used to establish a hypothesis of
chemical process for the connection between SST variations over the marginal seas of
East Asia and southern high latitude lower stratospheric ozone as follows: 1. The
warm (cold) SST anomalies over the marginal seas (Fig. 3) depress (enhance)
planetary wave activity in the middle to high latitudes of the Southern Hemisphere
(Figs 5 and 6). 2. The anomalous propagation of planetary waves into the stratosphere
and dissipation of ultra-long Rossby waves in the stratosphere strengthen/cool
(weaken/warm) the southern polar vortex (Fig. 7b and c). 3. A cooler (warmer) polar



vortex allows more (less) PSCs and active chlorine to form. 4. Consequently, southern
high latitude lower stratospheric ozone decreases (increases) (Fig. 7a).

3        However, it needs to point out that Antarctic polar vortex temperature is deeply

below the threshold for heterogeneous chemistry, so that a warming (cooling) in the
center of Antarctic polar vortex will have very little impact on Antarctic ozone by
affecting heterogeneous chemistry (Tilmes et al. 2006; Kirner et al. 2015). It seems to
challenge the above hypothesis.

8        Fig. 7c shows that the center of the correlation confidences locates near 60 °S. It

means that the center of stratospheric temperature changes caused by SST changes in
the East Asia Marginal Seas locates near 60 °S but not near 90 °S. Temperature change
near 60 °S maybe more effectively affects southern high latitude lower stratospheric
ozone than that near 90 °S since the background temperature in the lower stratosphere
near 60 °S would be higher than that near 90 °S. The chemical process maybe has a
certain contribution on the southern high latitude lower stratospheric ozone changes
caused by SST changes in the East Asia Marginal Seas.

16       We also found that the SST changes in the East Asia Marginal Seas are positively

correlated with southern high latitude stratospheric meridional wind (Fig. 7d),
suggesting a stronger (weaker) zonal circulation (Fig. 7b) related to the SST changes
impeding (promoting) transport of ozone from the middle latitude stratosphere to high
latitude stratosphere. Note that this correlation is the strongest in autumn but not in
summer when the south polar vortex is too stable that doesn't allow ozone rich air
into the vortex. Fig. 7d implies a dynamical contribution on the southern high latitude
lower stratospheric ozone changes caused by SST changes in the East Asia Marginal
Seas.

25       It is noteworthy that warm (cold) SST anomalies are generally thought to





increase (suppress) planetary wave activity via strengthening (weakening) convection
(Xie et al., 2008; Shu et al., 2010; Hu et al., 2014). However, this study shows that
warm (cold) SST anomalies over the marginal seas of East Asia suppress (increase)
planetary wave activity. This may be the warm (cold) SST anomalies over the
marginal seas in summer and autumn are equal to weaken (enhance) sea–land contrast
along the coastline of East Asia. This results in weaker (stronger) convection, which
suppresses (increases) planetary wave activity.
**4. Simulating the effect of SST changes in the marginal seas of East Asia on**
**southern high latitude lower stratospheric ozone**
We performed three time-slice simulations with WACCM4 to validate the mechanism
described in Section3. The monthly mean climatologies of surface emissions used in
the model were obtained from the A1B emissions scenario developed by the
Intergovernmental Panel on Climate Change (IPCC), and averaged over the period
1979–2015. QBO signals with a 28-month fixed cycle are included in WACCM4 as
an external forcing for zonal wind. The SST forcing used in the first time-slice
experiment (S1, the control experiment) was the 12-month climatology cycle
averaged over the period 1979–2015 and based on the HadISST dataset. S2 was a
sensitivity experiment and was the same as S1 except that warm anomalies (as in Fig.
3b) were added to the SST in the marginal seas of East Asia (5 ˚S–35 ˚N and 100–
140 ˚E). The third experiment, S3, was the same as S2, but with cold SST anomalies
(as in Fig. 3c). Detailed descriptions of experiments S1–S3 are provided in Table 2.
Figure 8 first shows the southern high latitude lower stratospheric ozone
anomalies forced by warm and cold SST anomalies over the marginal seas of East
Asia. It can be seen that the warm SST anomalies indeed cause ozone decrease in the



southern high latitude lower stratosphere (Fig. 8a) and cold SST anomalies results in
ozone increase (Fig. 8b). The simulations support the results shown from the
statistical analysis in Section 3.
Figure 9 shows the E–P flux vectors and divergence anomalies in the
stratosphere caused by SST anomalies over the marginal seas of East Asia. Analysis
of changes in the E–P flux (Eliassen and Palm 1961; Andrews et al. 1987) is often
used as a diagnostic for planetary wave propagation from the troposphere to the
stratosphere (Edmon et al., 1980). During periods of warm (cold) SST over the
marginal seas of East Asia, a decrease (increase) in upward wave flux entering the
stratosphere accompanied by stronger (weaker) convergence of the E–P flux in the
stratosphere at middle to high latitudes of the Southern Hemispheres (ca. 60°S) is
evident (Fig. 9a and c). The anomalous wave flux entering the stratosphere around
60°S confirms the result in Fig. 5, which shows that the wave rays terminate at about
60°S.
Many previous studies have demonstrated a strongly negative correlation
between upward propagating wave activity and the intensity of the stratospheric polar
vortex, with an anomalously negative and positive upward wave flux alongside a
stronger and weaker polar vortex, respectively (Christiansen 2001; Polvani and
Waugh 2004; Li and Lau 2013). During periods of warm (cold) SST over the marginal
seas of East Asia, the anomalous downward (upward) E–P flux, and larger (smaller)
E–P flux divergence at middle to high latitudes (ca. 60°S) in the Southern Hemisphere
(Fig. 9a and c), imply suppressed (active) wave activity in the stratosphere, which
induces a strengthened (weakened) circulation at southern polar vortex edge (Fig. 9b
and d). Finally, the cold (warm) polar vortex (Fig. 10a and c) allows more (less)





PSCs/active chlorine (Fig. 10b and d) to form. This is one process through which SST
variations over the marginal seas of East Asia causes southern high latitude lower
stratospheric ozone changes. The other process is that the strengthened (weakened)
southern polar vortex impedes (promotes) air exchange between middle and high
latitude stratosphere (Figure 11), and further decreases (increases) southern high
latitude lower stratospheric ozone levels.

7       As a result of human activity, the amount of Antarctic stratospheric ozone has

decreased remarkably over recent decades (Solomon 1990, 1999; Ravishankara et al.,
1994, 2009). At the same time, SST over the marginal seas of East Asia has followed
an increasing trend, but superimposed on strong interannual variations (Zheng et al.,
2014). Figure 12 shows the correlation coefficients between southern high latitude
lower stratospheric ozone and SST in which the SST and southern high latitude lower
stratospheric ozone variations have not been detrended as that in Fig. 2. Comparing
Fig. 12 with Fig. 2, we can see that the negative correlation coefficients over the
marginal seas of East Asia become larger in Fig. 12, implying a contribution of
warmer SST in the marginal seas of East Asia to decline trend of southern high
latitude lower stratospheric ozone.
We used ensemble transient experiments to estimate the contribution of SST
variations in the marginal seas of East Asia to southern high latitude lower
stratospheric ozone changes. The transient experiments incorporated the following
natural and anthropogenic external forcings for the period 1955–2005: observed SST
from the HadISST dataset, surface emissions from the IPCC A1B emissions scenario,
spectrally resolved solar variability (Lean et al., 2005), volcanic aerosols (from the
Stratospheric Processes and their Role in Climate (SPARC) Chemistry–Climate
Model Validation (CCMVal) REF-B2 scenario recommendations), and nudged QBO



(the time series in CESM is determined from the observed climatology). The first
transient experiment, T1, was the historical experiment covering the period 1955–
2005 (Marsh et al., 2013). The second transient experiment, T2, was the same as T1
except that the SST in the marginal seas of East Asia (5 °S–35 °N and 100–140 °E) for
the period 1955–2005 was replaced by the 12-month cycle of climatology averaged
over the same period. This means that in T2, the SST over the marginal seas of East
Asia had only a seasonal cycle, but no trend and no interannual variability. T3 was the
same as T2, but used a slightly different initial condition as an ensemble experiment.
Detailed descriptions of runs T1–T3 are provided in Table 3.
Figure 13a and b shows the southern high latitude lower stratospheric ozone
variations over the period 1955–2005 from T1 and the ensemble experiments
((T2+T3)/2). The southern high latitude lower stratospheric ozone variations caused
by the SST variability over the marginal seas of East Asia can be obtained by
subtracting simulated ozone in the ensemble experiments ((T2+T3)/2)) from the
ozone in T1 (Fig. 13c). There are evident differences in southern high latitude lower
stratospheric ozone variations between T1 and the ensemble experiments
((T2+T3)/2)). This illustrates that the SST variability over the marginal seas of East
Asia (Fig. 13d) does have a significant effect on southern high latitude lower
stratospheric ozone over the past five decades (Fig. 13c). The correlation coefficient
between the two lines in Fig. 13c and d is 0.29 which is significant at 99% confident
level. A further analysis reveals that the linear trend of ozone variations over the
region 200–50 hPa and 60–90 °S from T1 (Trend1, Fig. 3a) is $-1.2 \times e^{-3}$ ppmv/month,
and from (T1 – (T2+T3)/2) (Trend2, Fig. 3c) is $-0.204 \times e^{-3}$ ppmv/month. See Table 4.
It implies that the increasing linear trend in SST over the marginal seas of East Asia
can contribute approximately 17% of the declining trend in southern high latitude



lower stratospheric ozone from 1955–2005 (Trend2 / Trend1 $\times 100\%$).
**5. Conclusions and Summary**
In this study, the connection between SST and the southern high latitude lower
stratospheric ozone variations at the interannual time scale is examined. We found that
SST over the marginal seas of East Asia can significantly modulate the interannual
variability of southern high latitude lower stratospheric ozone and the processes
involved in this modulation are related to anomalous planetary wave activity induced
by SST variations over the marginal seas of East Asia. The planetary waves
originating from the marginal seas can propagate to the middle and high latitudes of
the Southern Hemisphere in summer and autumn via the North Australia–Southern
Hemisphere and South Africa–Southern Hemisphere pathways. The anomalous
propagation and dissipation of ultra-long Rossby waves in the stratosphere
strengthens/cools (weakens/warms) the southern polar vortex, which allows more
(less) active chlorine to form and deplete more (less) ozone on one hand. On the other
hand, a stronger (weaker) polar vortex impedes (promotes) the transport of middle
latitude ozone to high latitudes and further decreases (increases) southern high
latitude lower stratospheric ozone. The above results and analysis are based on
observations but are also supported by time-slice experiments conducted using the
CESM.

21        Our transient model simulations further demonstrated that SST variations over

the marginal seas of East Asia not only modulate the interannual variability of
southern high latitude lower stratospheric ozone, but also contribute to southern high
latitude lower stratospheric ozone trend over the past five decades. Our analysis
reveals that the trend of increasing SST over the marginal seas of East Asia may have





contributed approximately 17% to the decreasing trend of southern high latitude lower
stratospheric ozone over the past five decades.
**Acknowledgments.** Funding for this project is provided by the Science Foundations
of China (41575038, 41375072, 41575039, and 41530423) and 973 project of China
(2014CB441202). The SLIMCAT modelling work is supported by the UK National
Centre for Atmospheric Science (NCAS) and the CESM model is provide by NCAR.
We acknowledge the datasets from the ERA-interim and MERRA2, and the program
to calculate wave ray paths from http://ljp.gcess.cn/dct/page/65646.





**Rerferences**
Andrews, D. G., Holton, J. R., and Leovy, C. B.: Middle atmosphere dynamics, Academic press,
489 pp., 1987.
Angell, J. K.: Relation of Antarctic 100 mb temperature and total ozone to equatorial QBO,
equatorial SST, and sunspot number, 1958‐87, Geophys. Res. Lett., 15, 915–918, 1988.
Angell, J. K.: Influence of equatorial QBO and SST on polar total ozone, and the 1990 Antarctic
Ozone Hole, Geophys. Res. Lett., 17, 1569–1572, 1990.
Austin, J. and Wilson, R. J.: Ensemble simulations of the decline and recovery of stratospheric
ozone, J. Geophys. Res., 111, D16314, doi:10.1029/2005JD006907, 2006.
Austin, J., Scinocca, J., Plummer, D., Oman, L., Waugh, D., Akiyoshi, H., Bekki, S., Braesicke, P.,
Butchart, N., Chipperfield, M., Cugnet, D., Dameris, M., Dhomse, S., Eyring, V., Frith, S.,
Garcia, R. R., Garny, H., Gettelman, A., Hardiman, S. C., Kinnison, D., Lamarque, J. F.,
Mancini, E., Marchand, M., Michou, M., Morgenstern, O., Nakamura, T., Pawson, S., Pitari,
G., Pyle, J., Rozanov, E., Shepherd, T. G., Shibata, K., Teyssèdre, H., Wilson, R. J., and
Yamashita, Y.: Decline and recovery of total column ozone using a multimodel time series
analysis, J. Geophys. Res., 115, D00M10, doi:10.1029/2010JD013857, 2010.
Bodeker, G. E. and Scourfield, M. W. J.: Planetary waves in total ozone and their relation to
Antarctic ozone depletion, Geophys. Res. Lett., 22, 2949–2952, 1995.
Bowman, K. P.: Global Patterns of the Quasi-Biennial Oscillation in Total Ozone, J. Atmos. Sci.,
46, 3328–3343, 1989.
Bretherton, C. S., Widmann, M., Dymnikov, V. P., Wallace, J. M., and Bladé, I.: The Effective
Number of Spatial Degrees of Freedom of a Time-Varying Field, J. Climate, 12, 1990–2009,

23      1999.

Cagnazzo, C., Manzini, E., Calvo, N., Douglass, A., Akiyoshi, H., Bekki, S., Chipperfield, M.,
Dameris, M., Deushi, M., Fischer, A. M., Garny, H., Gettelman, A., Giorgetta, M. A.,
Plummer, D., Rozanov, E., Shepherd, T. G., Shibata, K., Stenke, A., Struthers, H., and Tian,
W.: Northern winter stratospheric temperature and ozone responses to ENSO inferred from
an ensemble of Chemistry Climate Models, Atmos. Chem. Phys., 9, 8935–8948, 2009.




Cagnazzo, C., Manzini, E., Fogli, P. G., Vichi, and M., Davini, P.: Role of Stratospheric
Dynamics in the Ozone-Carbon connection in the Southern Hemisphere, Clim. Dynam., 41,

3    3039-3054, 2013.

Calvo, N., Garcia, R., Garcia Herrera, R., Gallego, D., Gimeno, L., Hernández, E., and Ribera P.:
Analysis of the ENSO signal in tropospheric and stratospheric temperatures observed by
MSU, 1979– 2000, J. Climate, 17, 3934–3946, 2004.
Calvo, N., Giorgetta, M. A., Garcia-Herrera R., and Manzini, E.: Nonlinearity of the combined
warm ENSO and QBO effects on the Northern Hemisphere polar vortex in MAECHAM5
simulations, J. Geophys. Res., 114, D13109, doi:10.1029/2008JD011445, 2009.
Chandra, S. and Mcpeters, R. D.: The Solar-Cycle Variation of Ozone in the Stratosphere Inferred
from Nimbus-7 and Noaa-11 Satellites, J. Geophys. Res., 99, 20665–20671, 1994.
Chipperfield, M.: New version of the TOMCAT/SLIMCAT off‑line chemical transport model:
Intercomparison of stratospheric tracer experiments, Q. J. Roy. Meteor. Soc., 132, 1179–1203,

14    2006.

Christiansen, B.: Downward propagation of zonal mean zonal wind anomalies from the
stratosphere to the troposphere: Model and reanalysis, J. Geophys. Res., 106, 27307–27322,
doi:10.1029/2000jd000214, 2001.
Clem K. R., Renwick J. A., and McGregor J.: Relationship between eastern tropical Pacific
cooling and recent trends in the Southern Hemisphere zonal-mean circulation, Clim. Dyn., 1–

20    17, 2016.

Coy, L., Wargan, K., Molod, A., McCarty, W., and Pawson, S.: Structure and Dynamics of the
Quasi-Biennial Oscillation in MERRA-2, J. Climate, 29, 5339–5354, 2016.
Creilson, J. K., Fishman, J., and Wozniak, A. E.: Arctic Oscillation - induced variability in
satellite-derived tropospheric ozone, Geophys. Res. Lett., 32, L14822,
doi:10.1029/2005GL023016, 2005.
Dee, D. P., et al.: The ERA-Interim reanalysis: Configuration and performance of the data
assimilation system, Q. J. Roy. Meteor. Soc., 137, 553–597, 2011.
Dhomse, S. S., Weber, S. M., Wohltmann, I., Rex, M., and Burrows, J. P.: On the possible causes



of recent increases in northern hemispheric total ozone from a statistical analysis of satellite data from 1979 to 2003, Atmos. Chem. Phys., 6, 1165–1180, 2006.

Dhomse, S. S., Chipperfield, M. P., Feng, W., Hossaini, R., Mann, G. W., and Santee, M. L.: Revisiting the hemispheric asymmetry in midlatitude ozone changes following the Mount Pinatubo eruption: A 3–D model study, Geophys. Res. Lett., 42, 3038–3047, 2015.

Dhomse, S. S., Chipperfield, M. P., Damadeo, R. P., Zawodny, J. M., Ball, W. T., Feng, W., Hossaini, R., Mann, G. W., and Haigh J. D.: On the ambiguous nature of the 11–year solar cycle signal in upper stratospheric ozone, Geophys. Res. Lett., 43, 7241–7249, 2016.

Douglass, A. R., Strahan, S. E., Oman, L. D., and Stolarski, R. S.: Understanding differences in chemistry climate model projections of stratospheric ozone, J. Geophys. Res., 119, 4922–4939, 2014.

Edmon, H. J., Hoskins, B. J., and Mcintyre, M. E.: Eliassen-Palm cross-sections for the troposphere, J. Atmos. Sci., 37, 2600–2616, 1980.

Eliassen, A. and Palm, E.: On the transfer of energy in stationary mountain waves, Geofysiske Publikasjoner, 22, 1–23, 1961.

Evtushevsky, O. M., Kravchenko V. O., Hood L. L., Milinevsky G. P.: Teleconnection between the central tropical Pacific and the Antarctic stratosphere: spatial patterns and time lags, Clim. Dyn., 44, 1841–1855, 2015.

Eyring, V., et al., : Multi-model assessment of stratospheric ozone return dates and ozone recovery in CCMVal-2 models, Atmos. Chem. Phys., 10, 9451–9472, 2010.

Farman, J. G., Gardiner, B. G., and Shanklin, J. D.: Large losses of total ozone in Antarctica reveal seasonal ClOx/NOx interaction, Nature, 915, 207–210, 1985.

Feldstein, S. B.: Subtropical rainfall and the Antarctic ozone hole, Science, 332, 925–926, 2011.

Feng, W., Chipperfield, M. P., Davies, S., von der Gathen, P., Kyrö E., Volk, C. M., Ulanovsky, A., and Belyaev G.: Large chemical ozone loss in 2004/2005 Arctic winter/spring, Geophys. Res. Lett., 34, L09803, doi:10.1029/2006GL029098, 2007.

Feng, W., Chipperfield, M. P., Davies, S., Mann, G. W., Carslaw, K. S., Dhomse, S., Harvey, L., Randall, C., and Santee M. L.: Modelling the effect of denitrification on polar ozone



depletion for Arctic winter 2004/2005, Atmos. Chem. Phys., 11, 6559–6573, 2011.

Forster, P. and Shine, K.: Radiative forcing and temperature trends from stratospheric ozone changes, J. Geophys. Res., 102, 10841–10855, 1997.

Free, M. and Seidel, D. J.: The observed ENSO temperature signal in the stratosphere, J. Geophys. Res., doi:10.1029/2009JD012420, 2009.

Frossard, L., Rieder, H. E., Ribatet, M., Staehelin, J., Maeder, J. A., Di Rocco, S., Davison, A. C., and Peter, T.: On the relationship between total ozone and atmospheric dynamics and chemistry at mid-latitudes - Part 1: Statistical models and spatial fingerprints of atmospheric dynamics and chemistry, Atmos. Chem. Phys., 13, 147–164, 2013.

Fujiwara, M., Kita, K., and Ogawa, T.: Stratosphere-troposphere exchange of ozone associated with the equatorial Kelvin wave as observed with ozonesondes and rawinsondes, J. Geophys. Res., 103, 19173–19182, 1998.

Fusco, A. C. and Salby, M. L.: Interannual variations of total ozone and their relationship to variations of planetary wave activity, J. Climate, 12, 1619 – 1629, 1999.

Garcia, R. R. and Solomon, S. A.: possible relationship between interannual variability in Antarctic ozone and the quasi-biennial oscillation, Geophys. Res. Lett., 14, 848 –851, 1987.

Garcia, R. R., Marsh, D. R., Kinnison, D. E., Boville, B. A., and Sassi, F.: Simulation of secular trends in the middle atmosphere, 1950–2003, J. Geophys. Res., 112, D09301, doi:10.1029/2006JD007485, 2007.

García-Herrera, R., Calvo, N., Garcia, R. R., and Giorgetta, M. A.: Propagation of ENSO temperature signals into the middle atmosphere: A comparison of two general circulation models and ERA-40 reanalysis data, J. Geophys. Res., 111, D06101, doi:10.1029/2005JD006061, 2006.

Garfinkel, C. I. and Hartmann, D. L.: Effects of El Nino – Southern Oscillation and the Quasi-Biennial Oscillation on polar temperatures in the stratosphere, J. Geophys. Res., 112, D19112, doi:10.1029/2007JD008481, 2007.

Garfinkel, C. I. and Hartmann, D. L.: Different ENSO teleconnections and their effects on the stratospheric polar vortex, J. Geophys. Res., 113, D18114, doi:10.1029/2008JD009920,



1    2008.

Grassi, B., Redaelli G., and Visconti G.: Simulation of polar Antarctic trends: Influence of tropical

3        SST, Geophys. Res. Lett., 32, L23806, doi:10.1029/2005GL023804, 2005.

Grassi, B., Redaelli G., and Visconti G.: A physical mechanism of the atmospheric response over

5        Antarctica to decadal trends in tropical SST, Geophys. Res. Lett., 33, L17814,

6        doi:10.1029/2006GL026509, 2006.

Gettelman, A., Randel, W. J., Massie, S., and Wu, F.: El Niño as a Natural Experiment for

8        Studying the Tropical Tropopause Region, J. Climate, 14, 3375–3392, 2001.

Gray, L. J. and Ruth, S.: The Modeled Latitudinal Distribution of the Ozone Quasi-Biennial

10       Oscillation Using Observed Equatorial Winds, J. Atmos. Sci., 50, 1033–1046, 1993.

Hadjinicolaou, P., Pyle, J. A., Chipperfield, M. P., and Kettleborough, J. A.: Effect of interannual

12       meteorological variability on mid-latitude $O_3$, Geophys. Res. Lett., 24, 2993–2996, 1997.

Hadjinicolaou, P., Jrrar, A., Pyle, J. A., and Bishop, L.: The dynamically driven long-term trend in

14       stratospheric ozone over northern middle latitudes, Q. J. Roy. Meteor. Soc., 128, 1393–1412,

15       2002.

Hadjinicolaou, P., and Pyle, J. A.: The Impact of Arctic Ozone Depletion on Northern Middle

17       Latitudes: Interannual Variability and Dynamical Control, J. Atmos. Chem., 47, 25–43, 2004.

Hess, P. G. and Lamarque, J. F.: Ozone source attribution and its modulation by the Arctic

19       oscillation during the spring months, J. Geophys. Res., 112, D11303,

20       doi:10.1029/2006JD007557, 2007.

Hoerling, M. P., Hurrell, J. W., and Xu, T. Y.: Tropical origins for recent North Atlantic climate

22       change, Science, 292, 90–92, doi:10.1126/science.1058582, 2001.

Hoerling, M. P., Hurrell, J. W., Xu, T., Bates, G. T., and Phillips, A. S.: Twentieth century North

24       Atlantic climate change. Part II: Understanding the effect of Indian Ocean warming, Clim.

25       Dynam., 23, 391–405, doi:10.1007/s00382-004-0433-x, 2004.

Hofmann, D. J. and Oltmans, S. J.: Anomalous Antarctic Ozone during 1992 - Evidence for

27       Pinatubo Volcanic Aerosol Effects, J. Geophys. Res., 98, 18555–18561, 1993.

Hu, Y. and Fu, Q.: Stratospheric warming in Southern Hemisphere high latitudes since 1979,




Atmos. Chem. Phys., 9, 4329–4340, 2009.
Hu, Y., and Pan L.: Arctic stratospheric winter warming forced by observed SST, Geophys. Res.
Lett., 36, L11707, doi:10.1029/2009GL037832, 2009.
Hu, Y. and Pan, L.: Arctic stratospheric winter warming forced by observed SST, Geophys. Res.
Lett., 36, L11707, doi:10.1029/2009GL037832, 2009.
Hu, D., Tian, W., Xie, F., Shu, J., and Dhomse, S.: Effects of meridional sea surface temperature
changes on the stratospheric temperature and circulation, Adv. Atmos. Sci., 31, 888–900,
doi:10.1007/s00376-013-3152-6, 2014.
Huck, P. E., McDonald, A. J., Bodeker, G. E., and Struthers, H.: Interannual variability in
Antarctic ozone depletion controlled by planetary waves and polar temperature, Geophys.
Res. Lett., 32, 370–370, 2005.
Hurwitz, M. M., Newman, P. A., Oman, L. D., and Molod, A. M.: Response of the Antarctic
Stratosphere to Two Types of El Niño Events, J. Atmos. Sci., 68, 812-822,
doi:10.1175/2011JAS3606.1, 2011a.
Hurwitz, M. M, Song, I. S., Oman, L. D., Newman, P. A., Molod, A. M., Frith, S. M., and Nielsen,
J. E.: Response of the Antarctic stratosphere to warm pool El Niño Events in the GEOS CCM,
Atmos. Chem. Phys., 11, 9659–9669, doi:10.5194/acp-11-9659-2011, 2011b.
Kang, S. M., Polvani, L. M., Fyfe, J. C., and Sigmond, M.: Impact of polar ozone depletion on
subtropical precipitation, Science, 332, 951–954, 2011.
Keeble, J., Braesicke, P., Abraham, N. L., Roscoe, H. K., and Pyle, J. A.: The impact of polar
stratospheric ozone loss on Southern Hemisphere stratospheric circulation and climate,
Atmos. Chem. Phys., 14, 13705–13717, 2014.
Kerrj, B. and Mcelroy, C. T.: Evidence for large upward trends of ultraviolet-B radiation linked to
ozone depletion, Science, 262, 1032–1034, 1993.
Kirner, O., Müller, R., Ruhnke, R., and Fischer, H.: Contribution of liquid, NAT and ice particles
to chlorine activation and ozone depletion in Antarctic winter and spring, Atmos. Chem.
Phys., 15, 2019-2030, 2015.
Lait, L. R., Schoeberl, M. R., and Newman, P. A.: Quasi-biennial modulation of the Antarctic



ozone depletion, J. Geophys. Res., 94, 11559–11571, 1989.

Lamarque, J. F. and Hess, P. G.: Arctic Oscillation modulation of the Northern Hemisphere spring tropospheric ozone, Geophys. Res. Lett., 31, L06127, doi:10.1029/2003GL019116, 2004.

Lean, J., Rottman, G., Harder, J., and Kopp, G.: SORCE contributions to new understanding of global change and solar variability, Sol. Phys., 230, 27–53, 2005.

Li, F., Vikhliaev, Y. V., Newman, P. A., Pawson, S., Perlwitz, J., Waugh, D. W., and Douglass, A. R.: Impacts of Interactive Stratospheric Chemistry on Antarctic and Southern Ocean Climate Change in the Goddard Earth Observing System, Version 5 (GEOS-5), J. Climate, 29, 3199–3218, 2016.

Li, K. F. and Tung, K. K.: Quasi-biennial oscillation and solar cycle influences on winter Arctic total ozone, J. Geophys. Res., 119, 5823–5835, 2014.

Li, K. F., Tian, B., Waliser, D. E., Schwartz, M. J., Neu, J. L., Worden, J. R., and Yung, Y. L.: Vertical structure of MJO-related subtropical ozone variations from MLS, TES, and SHADOZ data, Atmos. Chem. Phys., 12, 425–436, 2012.

Li, Y. J., Li, J., Jin, F. F., and Zhao, S.: Interhemispheric Propagation of Stationary Rossby Waves in a Horizontally Nonuniform Background Flow, J. Atmos. Sci., 72, 3233–3256, 2015.

Li, Y. and Lau, N. C.: Influences of ENSO on stratospheric variability, and the descent of stratospheric perturbations into the lower troposphere, J. Climate, 26, 4725–4748, 2013.

Li, S. L., Perlwitz, J., Hoerling, M. P., and Chen, X. T.: Opposite Annular Responses of the Northern and Southern Hemispheres to Indian Ocean Warming, J. Climate, 23, 3720–3738, 2010.

Li, S. L. and Chen, X. T.: Quantifying the Response Strength of the Southern Stratospheric Polar Vortex to Indian Ocean Warming in Austral Summer, Adv. Atmos. Sci., 31, 492–503, 2014.

Lin, P., Fu, Q., and Hartmann, D.: Impact of tropical SST on stratospheric planetary waves in the Southern Hemisphere, J. Climate, 25, 5030-5046, doi:http://dx.doi.org/10.1175/JCLI-D-11-00378.1, 2012.

Li, Y., Li, J., and Feng, J. A.: Teleconnection between the Reduction of Rainfall in Southwest Western Australia and North China, J. Climate, 25, 8444-8461, 2012.





Liu, C. X., Liu, Y., Cai, Z. N., Gao, S. T., Lu, D. R., and Kyrola, E.: A Madden-Julian
Oscillation-triggered record ozone minimum over the Tibetan Plateau in December 2003 and
its association with stratospheric "low-ozone pockets", Geophys. Res. Lett., 36, L15830,
doi:10.1029/2009GL039025, 2009.
Liu, J. J., Jones, D. B. A., Zhang, S., and Kar, J.: Influence of interannual variations in transport on
summertime abundances of ozone over the Middle East, J. Geophys. Res., 116, D20310,
doi:10.1029/2011JD016188, 2011.
Liu, J., Tarasick, D. W., Fioletov, V. E., McLinden, C., Zhao, T., Gong, S., Sioris, C., Jin, J. J., Liu,
G., and Moeini O.: A global ozone climatology from ozone soundings via trajectory mapping:
a stratospheric perspective, Atmos. Chem. Phys., 13, 11441–11464, 2013.
Mancini, E., Visconti, G., Pitart, G., and Verdecch, M.: An estimate of the Antarctic ozone
modulation by the QBO, Geophys. Res. Lett., 18, 175-178, 1991.
Manzini, E., Giorgetta, M. A., Esch, M., Kornblueh, L., and Roeckner, E.: The Influence of Sea
Surface Temperatures on the Northern Winter Stratosphere: Ensemble Simulations with the
MAECHAM5 Model, J. Climate, 19, 3863–3881, 2006.
Manney, G., Zurek, R., O'Neill, A., and Swinbank, R.: On the Motion of Air through the
Stratospheric Polar Vortex. J. Atmos. Sci., 51, 2973–2994, 1994.
Marsh, D. R., Mills, M. J., Kinnison, D. E., Lamarque, J.-F., Calvo, N., and Polvani, L. M.:
Climate change from 1850 to 2005 simulated in CESM1 (WACCM), J. Climate, 26, 7372–73
20    91, 2013.

Müller, R., Peter, T., Crutzen, P. J., Oelhaf, H., Adrian, G. P., Von Clarmann, T., Wegner, A.,
Schmidt, U., and Lary, D.: Chlorine chemistry and the potential for ozone depletion in the
arctic stratosphere in the winter of 1991/92, Geophys. Res. Lett., 21, 1427–1430, 1994.
Müller, R., Tilmes, S., Konopka, P., Grooß, J.-U., and Jost H.-J.: Impact of mixing and chemical
change on ozone-tracer relations in the polar vortex, Atmos. Chem. Phys., 5, 3139–3151,
26    2005.

Perlwitz, J., Pawson, S., Fogt, R. L., Nielsen, J. E., and Neff, W. D.: Impact of stratospheric ozone
hole    recovery    on    Antarctic    climate,    Geophys.    Res.    Lett.,    35,    L08714,





doi:10.1029/2008GL033317, 2008.

Polvani, L. M. and Waugh, D. W.: Upward wave activity flux as a precursor to extreme stratospheric events and subsequent anomalous surface weather regimes, J. Climate, 17, 3548–3554, 2004.

Polvani, L. M., Waugh, D. W., Correa, G. J. P., and Son, S.-W.: Stratospheric ozone depletion: The main driver of twentieth-century atmospheric circulation changes in the Southern Hemisphere, J. Climate, 24, 795-812, doi:10.1175/2010JCLI3772.1, 2011.

Prather, M. J.: Numerical advection by conservation of second-order moments, J. Geophys. Res., 91, 6671–6681, 1986.

Previdi, M. and Polvani, L. M.: Climate system response to stratospheric ozone depletion and recovery, Q. J. Roy. Meteor. Soc., 140, 2401-2419, doi:10.1002/qj.2330, 2014.

Randel, W. J., Garcia, R. R., Calvo, N., and Marsh, D.: ENSO influence on zonal mean temperature and ozone in the tropical lower stratosphere, Geophys. Res. Lett., 36, L15822, doi:10.1029/2009GL039343, 2009.

Rao, J., and Ren R., A decomposition of ENSO's impacts on the northern winter stratosphere: competing effect of SST forcing in the tropical Indian Ocean, *Clim. Dyn.*, 1–19, doi:10.1007/s00382-015-2797-5, 2015.

Rayner, N. A., Parker,D. E., Horton, E. B., Folland, C. K., Alexander, L. V., Rowell, D. P., Kent, E. C., and Kaplan, A.: Global analysis of sea surface temperature, sea ice, and night marine air temperature since the late nineteenth century, J. Geophys. Res., 108, doi:10.1029/2002JD002670, 2003.

Ren, R. C., Cai M., Xiang C. Y., and Wu G. X.: Observational evidence of the delayed response of stratospheric polar vortex variability to ENSO SST anomalies, *Clim. Dyn.*, 38, 1345–1358, doi:10.1007/s00382-011-1137-7, 2012.

Rieder, H. E., Frossard, L., Ribatet, M., Staehelin, J., Maeder, J. A., Di Rocco, S., Davison, A. C., Peter, T., Weihs, P., and Holawe F.: On the relationship between total ozone and atmospheric dynamics and chemistry at mid-latitudes - Part 2: The effects of the El Nino/Southern Oscillation, volcanic eruptions and contributions of atmospheric dynamics and chemistry to



long-term total ozone changes, Atmos. Chem. Phys., 13, 165–179, 2013.
Rieder, H. E., Polvani, L. M., and Solomon, S.: Distinguishing the impacts of ozone depleting

3       substances and well-mixed greenhouse gases on Arctic stratospheric ozone and temperature

4       trends, Geophys. Res. Lett., 41, 2652–2660, 2014.

Rienecker, M. M., et al.: MERRA: NASA's modern-era retrospective analysis for research and

6       applications, J. Climate, 24, 3624–3648, 2011.

Rozanov, E. V., Schlesinger, M. E., Andronova, N. G., Yang, F., Malyshev, S. L., Zubov, V. A.,

8       Egorova, T. A., and Li, B.: Climate/chemistry effects of the Pinatubo volcanic eruption

9       simulated by the UIUC stratosphere/troposphere GCM with interactive photochemistry, J.

Geophys. Res., 107, 4594, doi:10.1029/2001JD000974, 2002.
Rozanov, E. V., Schraner, M., Egorova, T., Ohmura, A., Wild, M., Schmutz, W., and Peter, T.:
Solar signal in atmospheric ozone, temperature and dynamics simulated with CCM SOCOL
in transient mode, Memor. Soc. Astronom. Ital., 76, 876-879, 2005.
Salby, M. L. and Callaghan, P. F.: Systematic Changes of Northern Hemisphere Ozone and Their
Relationship to Random Interannual Changes, J. Climate, 17, 4512–4521, 2004.
Salby, M. L. and Callaghan, P. F.: Influence of planetary wave activity on the stratospheric final
warming and spring ozone, J. Geophys. Res., 112, 365-371, 2007a.
Salby, M. L. and Callaghan, P. F.: On the wintertime increase of Arctic ozone: Relationship to
changes of the polar-night vortex, J. Geophys. Res., 112, 541-553, 2007b.
Sassi, F., Kinnison, D., Boville, B. A., Garcia, R. R., and Roble, R.: Effect of El Niño-Southern
Oscillation on the dynamical, thermal, and chemical structure of the middle atmosphere, J.
Geophys. Res., 109, D17108, doi:10.1029/ 2003JD004434, 2004.
Schnadt, C., and Dameris M.: Relationship between North Atlantic Oscillation changes and
stratospheric ozone recovery in the Northern Hemisphere in a chemistry-climate model,
Geophys. Res. Lett., 30, 1487, doi:10.1029/ 2003GL017006, 2003.
Schoeberl, M. R. and Hartmann, D. L.: The dynamics of the stratospheric polar vortex and its
relation to springtime ozone depletions, Science, 251, 46–52, 1991.
Shindell, D. T. and Schmidt, G. A.: Southern Hemisphere climate response to ozone changes and





greenhouse gas increases, Geophys. Res. Lett., 31, L18209, doi:10.1029/2004GL020724,

2    2004.

Shindell, D. T., Wong, S., and Rind, D.: Interannual Variability of the Antarctic Ozone Hole in a
GCM. Part I: The Influence of Tropospheric Wave Variability, J. Atmos. Sci., 54, 2308-2319,

5    1997.

Shindell, D. T., Rind, D., and Balachandran, N.: Interannual Variability of the Antarctic Ozone
Hole in a GCM. Part II: A Comparison of Unforced and QBO-Induced Variability, J. Atmos.
Sci., 56, 1873–1884, 2010.
Shu, J., Tian, W., Hu, D., Zhang, J., Shang, L., Tian, H., and Xie, F.: Effects of the Quasi-biennial
Oscillation and Stratospheric Semiannual Oscillation on Tracer Transport in the upper
Stratosphere, J. Atmos. Sci., 70, 1370–1389, doi:10.1175/JAS-D-12-053.1, 2013.
Sigmond, M. and Fyfe, J. C.: The Antarctic Sea Ice Response to the Ozone Hole in Climate
Models, J. Climate, 27, 1336–1342, 2014.
Solomon, S.: Antarctic ozone: progress towards a quantitative understanding, Nature, 347, 347–

15    354, 1990.

Solomon, S.: Stratospheric ozone depletion: A review of concepts and history, Rev. Geophys., 37,
275–316, 1999.
Son, S.-W., Polvani, L. M., Waugh, D. W., Akiyoshi, H., Garcia, R., Kinnison, D., Pawson, S.,
Rozanov, E., Shepherd, T. G., and Shibata, K.: The impact of stratospheric ozone recovery on
the Southern Hemisphere westerly jet, Science, 320, 1486–1489, 2008.
Son, S.-W., Tandon, N. F., Polvani, L. M., and Waugh, D. W.: Ozone hole and Southern
Hemisphere climate change, Geophys. Res. Lett., 36, L15705, doi:10.1029/2009GL038671,

23    2009.

Son, S.-W., et al.: Impact of stratospheric ozone on Southern Hemisphere circulation change: A
multimodel assessment, J. Geophys. Res., 115, D00M07, doi:10.1029/2010JD014271, 2010.
Steinbrecht, W., Kohler U., Claude H., Weber M., Burrows J. P., and van der A, R. J.: Very high
ozone columns at northern mid-latitudes in 2010, Geophys. Res. Lett., 38, L06803,
doi:10.1029/2010GL046634, 2011.



Thompson, D. W. J., Solomon, S., Kushner, P. J., England, M. H., Grise, K. M., and Karoly, D. J.: Signatures of the Antarctic ozone hole in Southern Hemisphere surface climate change, Nature Geosci., 4, 741–749, 2011.

Tian, W. and Chipperfield, M. P.: A new coupled chemistry–climate model for the stratosphere: The importance of coupling for future O3-climate predictions, Q. J. Roy. Meteor. Soc., 131, 281–303, 2005.

Tian, B. J., Yung, Y. L., Waliser, D. E., Tyranowski, T., Kuai, L., Fetzer, E. J., and Irion, F. W.: Intraseasonal variations of the tropical total ozone and their connection to the Madden-Julian Oscillation, Geophys. Res. Lett., 34, L08704, doi:10.1029/2007GL029451, 2007.

Tilmes, S., Müller, R., Engel, A., Rex, M., and Russell III J. M.: Chemical ozone loss in the Arctic and Antarctic stratosphere between 1992 and 2005, Geophys. Res. Lett., 33, L20812, 2006.

Trenberth, K. E: The definition of El Niño, Bull. Am. Meteorol. Soc., 78, 2771–2777, 1997.

Tung, K. K. and Yang, H.: Dynamic variability of column ozone, J. Geophys. Res., 93, 11123–11128, 1988.

Weare, B. C.: Madden-Julian Oscillation in the tropical stratosphere, J. Geophys. Res., 115, D17113, doi:10.1029/2009JD013748, 2010.

Weber, M., Dhomse, S., Wittrock, F., Richter, A., Sinnhuber, B.-M., and Burrows, J. P.: Dynamical Control of NH and SH Winter/Spring Total Ozone from GOME Observations in 1995 – 2002, Geophys. Res. Lett., 30, 389–401, 2003.

Weiss, A. K., Staehelin, J., Appenzeller, C., and Harris, N. R. P.: Chemical and dynamical contributions to ozone profile trends of the Payerne (Switzerland) balloon soundings, J. Geophys. Res., 106, 22685–22694, 2001.

Welhouse, L. J., Lazzara M. A., Keller L. M., Tripoli G. J.,Hitchman M. H.: Composite analysis of the effects of ENSO events on Antarctica, J. Climate, 29, 1797–1808, 2016.

Wilson, A. B., Bromwich D. H., Hines K. M., Wang S.: El Niño Flavors and Their Simulated Impacts on Atmospheric Circulation in the High Southern Latitudes, J. Climate, 27, 8934–8955, 2014.

Yang, C, Li T, Dou X, Xue X.: Signal of central Pacific El Niño in the Southern Hemispheric



stratosphere during austral spring, J. Geophys. Res., 120, 2015.

Yu, J. Y., Paek H., Saltzman E. S., Lee T.: The Early 1990s Change in ENSO–PSA–SAM Relationships and Its Impact on Southern Hemisphere Climate, J. Climate, 28, 9393–9408, 2015.

Xie, F., Tian, W., and Chipperfield, M. P.: Radiative effect of ozone change on stratosphere-troposphere exchange, J. Geophys. Res., 113, D00B09, doi:10.1029/2008JD009 829, 2008.

Xie, F., Li, J., Tian, W., Feng, J., and Huo, Y.: The Signals of El Niño Modoki in the Tropical Tropopause Layer and Stratosphere, Atmos. Chem. Phys., 12, 5259–5273, doi:10.5194/acp-12-5259-2012, 2012.

Xie, F., Li, J., Tian, W., Zhang, J., and Shu, J.: The impacts of two types of El Nino on global ozone variations in the last three decades, Adv. Atmos. Sci., 31, 1113–1126, 2014a.

Xie, F., Li, J., Tian, W., Zhang, J., and Sun, C.: The relative impacts of El Nino Modoki, canonical El Nino, and QBO on tropical ozone changes since the 1980s, Environ. Res. Lett., 9, 064020, 2014b.

Xie F., Li, J., Tian, W., Fu, Q., Jin, F-F., Hu, Y., Zhang, J., Wang, W., Sun, C., Feng, J., Yang Y., and Ding, R.: A connection from Arctic stratospheric ozone to El Niño-Southern oscillation, Environ. Res. Lett., 11, 124026, 2016.

Zhao, S., Li, J., and Li, Y. J.: Dynamics of an Interhemispheric Teleconnection across the Critical Latitude through a Southerly Duct during Boreal Winter. J. Climate, 28, 7437–7456, 2015.

Zhao, S., Li, J., Li, Y., and Zheng, J.: Interhemispheric influence of the Indo-Pacific convection oscillation on Southern Hemisphere rainfall, Submitted to Climate Dynamics, 2016.

Zheng, J. Y., Li, J., and Feng, J.: A dipole pattern in the Indian and Pacific oceans and its relationship with the East Asian summer monsoon, Environ. Res. Lett., 9, 074006, doi:10.1088/1748-9326/9/7/074006, 2014.

Zhang, J., Tian, W., Xie, F., Tian, H., Luo, J., Zhang, J., Liu. W., and Dhomse, S.: Climate warming and decreasing total column ozone over the Tibetan Plateau during winter and spring, Tellus, 66B, 136–140, 2014.



1   Zhang, J., Tian, W. S., Wang, Z. W., Xie, F., and Wang, F. Y.: The Influence of ENSO on Northern

2       Midlatitude Ozone during the Winter to Spring Transition, J. Climate, 28, 4774–4793, 2015a.

3   Zhang, J., Tian, W. S., Xie, F., Li, Y. P., Wang, F. Y., Huang, J. L., and Tian, H. Y.: 2015b:

4       Influence of the El Niño southern oscillation on the total ozone column and clear-sky

5       ultraviolet radiation over China, Atmos. Environ., 120, 205–216, 2015b.

6   Zubiaurre, I. and Calvo, N.: The El Niño–Southern Oscillation (ENSO) Modoki signal in the

7       stratosphere, J. Geophys. Res., 117, D04104, doi:10.1029/2011JD016690, 2012.





Table 1. Warm and cold SST events in the marginal seas of East Asia from 1979 to 2015 analyzed
in this paper using the ST_MSEA index (Fig. 3a).

| Warm Events[*] | Cold Events[*] |
|---|---|
| JUN1983–NOV1983 | SEP1979–MAY1980 |
| MAY1987–NOV1988 | OCT1981–NOV1982 |
| NOV1997–MAR2000 | MAY1985–MAY1986 |
| MAR2002–AUG2003 | AUG1992–MAY1993 |
| AUG2008–FEB2009 | JUL2004–DEC2004 |
| MAY2010–NOV2010 | FEB2011–SEP2011 |
|  | MAY2012–APR2012 |
|  | NOV2014–SEP2015 |

[*]Following the definition of ENSO events (Trenberth 1997), we propose a threshold of ±0.15,
which is equal to the standard deviation of the ST_MSEA series, as the indicator of warm and cold
events.





1    **Table 2.** Experiments S1–S3.

| Experiments[*1] | Descriptions |
| --- | --- |
| S1 | Time-slice run using case F_2000_WACCM in CESM. The SST is the 12-month cycle climatology mean for the period 1979–2015 based on HadISST dataset (Rayner et al., 2003); the monthly mean climatologies of surface emissions used in the model are obtained from the A1B emissions scenario developed by the IPCC, averaged over the period 1979–2015. QBO phase signals with a 28-month fixed cycle are included in WACCM4 as an external forcing for zonal wind. |
| S2 | Same as S1, except that the SST in the marginal seas of East Asia (5°S–35°N and 100–140°E) adds warm SST anomalies (as Fig. 3b). |
| S3 | Same as S1, except that the SST in the marginal seas of East Asia (5°S–35°N and 100–140°E) adds cold SST anomalies (as Fig. 3c). |

2    [*1]Each experiment is run for 53 years, with the first 3 years excluded as a spin-up period. The

3    remaining 50 years are used for the analysis.





**Table 3.** Experiments T1–T3.

| Experiments[*1] | Descriptions |
|---|---|
| T1 | Transient run using case F_1955-2005_WACCM_CN in CESM. SST forcing based on HadISST dataset, surface emissions are obtained from the A1B emissions scenario developed by the IPCC, spectrally resolved solar variability (Lean et al., 2005), volcanic aerosols (from the SPARC CCMVal REF-B2 scenario recommendations), nudged QBO (the time series in CESM is determined from the observed climatology). |
| T2 | Same as T1, except that the SST in the marginal seas of East Asia (5 °S–35 °N and 100–140 °E) between 1955 and 2005 is replaced by the 12 months cycle of climatology averaged for the period 1955–2005. |
| T3 | Same as T2, but with slightly different initial condition[*2] |

[*1]Integration period is 1955–2005 for T1–T3.
[*2]The parameter \<pertlim\> is used to produce different initial conditions in the CESM model,
which produces an initial temperature perturbation. The magnitude was about $e^{-14}$.
Table 4. Linear trends of ozone variations over the region 200–50 hPa and 60–90 °S from
experiments with (T1) and without SST (T2 +T3) variations in the East Asia Marginal Seas (T1–3
see Table 3).

| Experiments | Values |
|---|---|
| Linear trend of ozone variations over the region 200–50 hPa and 60–90 °S from T1 (Trend1) | $-1.2 \times e^{-3}$ ppmv/month |
| Linear trend of ozone variations over the region 200–50 hPa and 60–90 °S from (T1 – (T2+T3)/2) (Trend2) | $-0.204 \times e^{-3}$ ppmv/month |



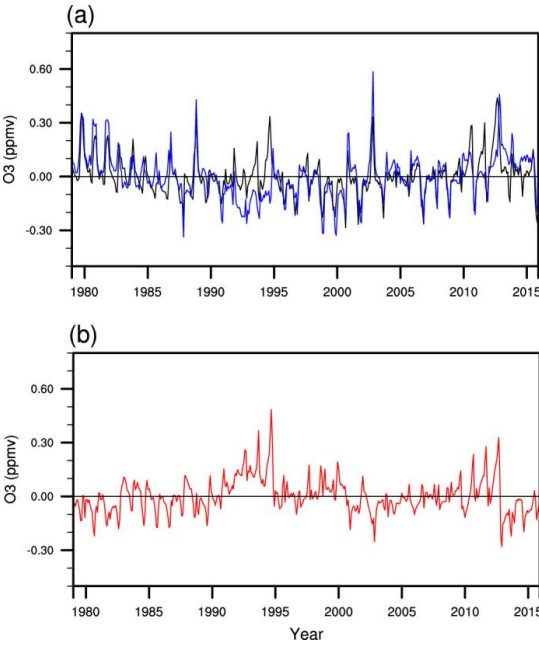

2 **Figure 1.** (a) Time series of southern high latitude lower stratospheric ozone variations averaged

3 over the region 60–90 ̊S at 200–50 hPa from the MERRA2 (black line), and SLIMCAT monthly

4 ozone (blue line) datasets. (b) The difference between MERRA2 and SLIMCAT ozone.





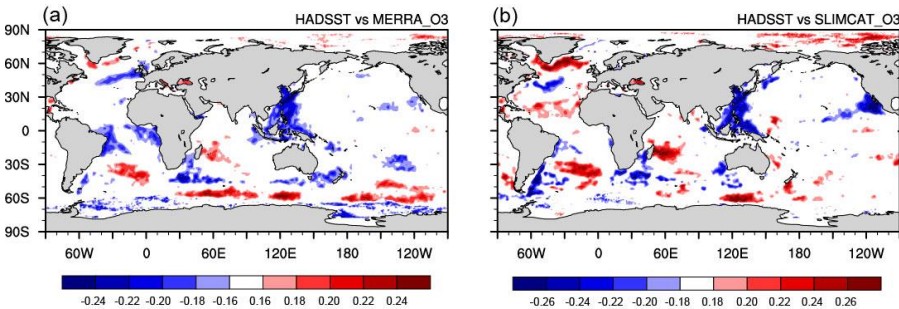

**Figure 2.** Correlation coefficients between southern high latitude lower stratospheric ozone
variations and SST (from HadISST) between 1979 and 2015. Southern high latitude lower
stratospheric ozone variations are averaged over the region 60–90 °S at 200–50 hPa. (a) Ozone
from MERRA2. (b) Ozone from SLIMCAT. Only the significant correlations are colored;
statistical significance was calculated using the two-tailed Student's $t$-test and the $N^{eff}$ of DOF (see
section 2). The seasonal cycles and linear trends were removed prior to calculating the correlation
coefficients.



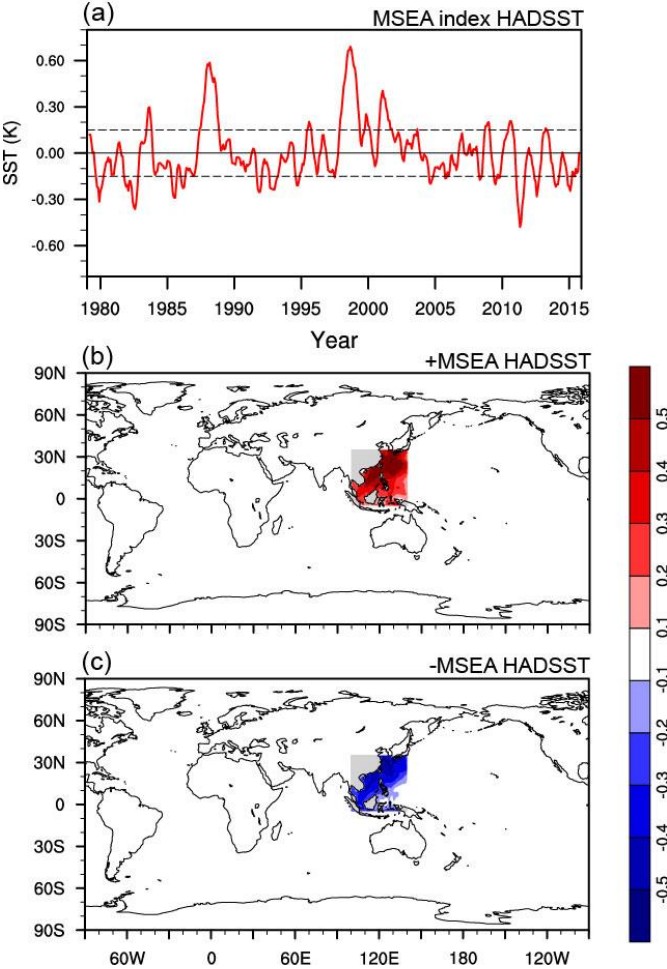

**Figure 3.** (a) SST variations in the marginal seas of East Asia defined using the ST_MSEA index
that was calculated by averaging SST over the region from 5˚S–35˚N at 100˚E–140˚E (from
HadISST), and then removing the seasonal cycle and linear trend. The dashed lines indicate the
thresholds for definition of warm and cold events. (b) and (c) show the composite warm and cold
SST anomalies, respectively, for the events listed in Table 1.



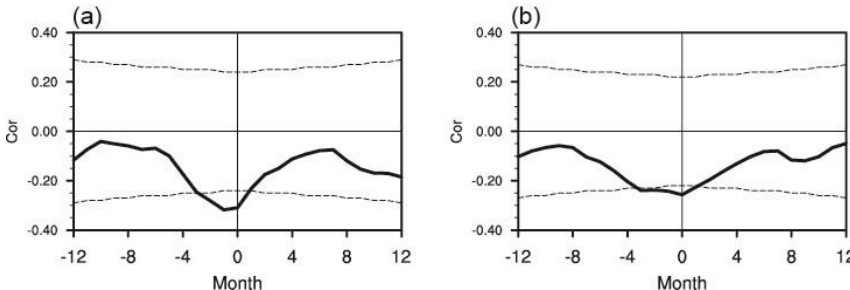

**Figure 4.** Lead–lag correlations between the ST_MSEA index and southern high latitude lower
stratospheric ozone variations between 1979 and 2015; Southern high latitude lower stratospheric
ozone variations are averaged over the region 60–90 °S at 200–50 hPa from the (a) MERRA2 and
(b) SLIMCAT datasets. Negative months on the *x*-axis refer to ST_MSEA leading southern high
latitude lower stratospheric ozone variations, and positive months refer to the southern high
latitude lower stratospheric ozone variations leading ST_MSEA. Dotted lines indicate the 90%
confidence level; lead–lag correlations exceeding the dotted lines are statistically significant. The
statistical significance of the lead–lag correlation between two auto-correlated time series was
calculated using the two-tailed Student's *t*-test and the $N^{eff}$ of DOF (see section 2).



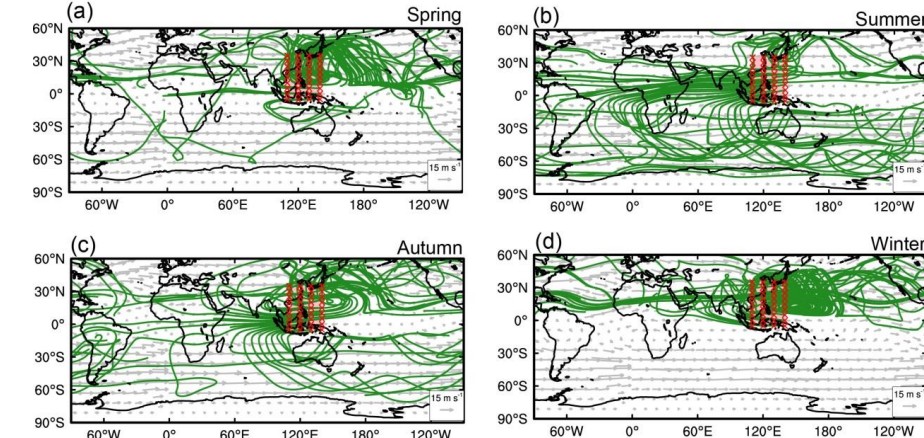

**Figure 5.** Ray paths (green lines) at 300 hPa in (a) spring, (b) summer, (c) autumn, and (d) winter.
Red points denote wave sources in the marginal seas of East Asia (5 °S–35 °N, 100 °E–140 °E). The
wavenumbers along these rays are in the range 1–5. The grey vectors indicate climatological
flows.





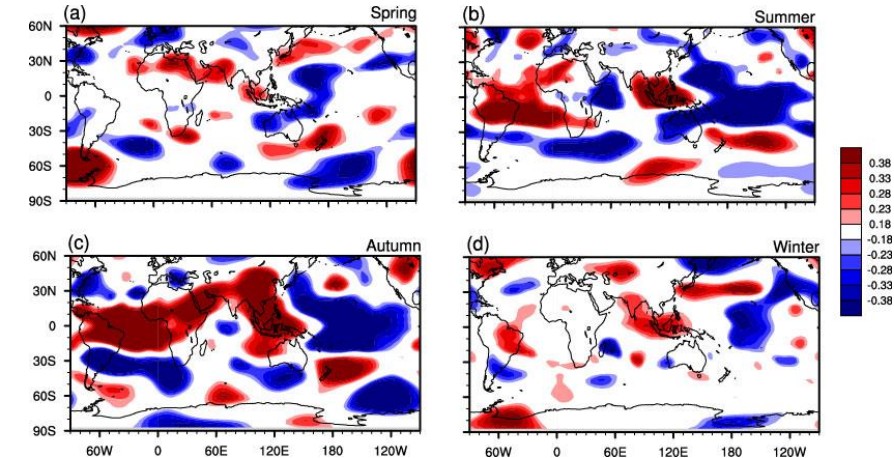

**Figure 6.** Correlation coefficients between the ST_MSEA index and 300-hPa geopotential height
from the ERA-Interim reanalysis in (a) spring, (b) summer, (c) autumn, and (d) winter between
1979 and 2015. Only significant correlations are colored. The seasonal cycles and linear trends
were removed before calculating the correlation coefficients.





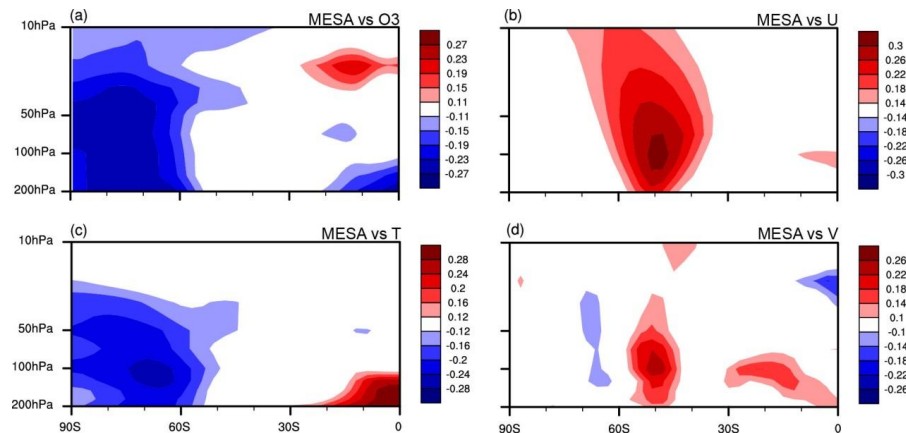

**Figure 7.** Correlation coefficients between ST_MSEA and (a) zonally averaged ozone, (b) zonal
wind, (c) temperature, and (d) meridional wind. Wind and temperature from ERA-Interim
reanalysis data; ozone from MERRA2. Only significant correlations are colored. The seasonal
cycles and linear trends were removed before calculating the correlation coefficients.





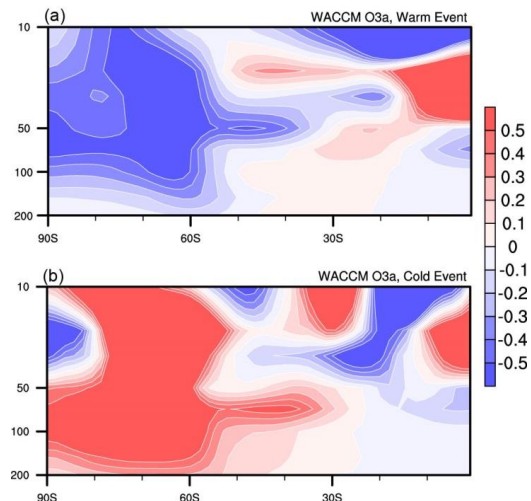

Figure 8. Zonal mean differences in ozone (ppmv) between WACCM simulations (a) S2 and S1, and (b) S3 and S1.



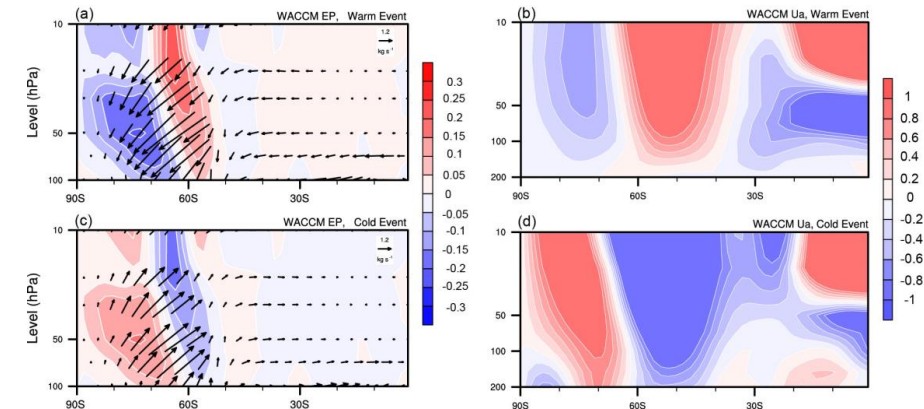

2 **Figure 9.** Differences in E–P flux vectors (black arrows) and divergence (color shading) between

3 (a) S2 and S1, and (c) S3 and S1. Units for the horizontal and vertical vector directions are $10^7$ and

4 $10^5$ kg s$^{-1}$, respectively. (b) and (d), as (a) and (c), but for zonal wind (m s$^{-1}$).





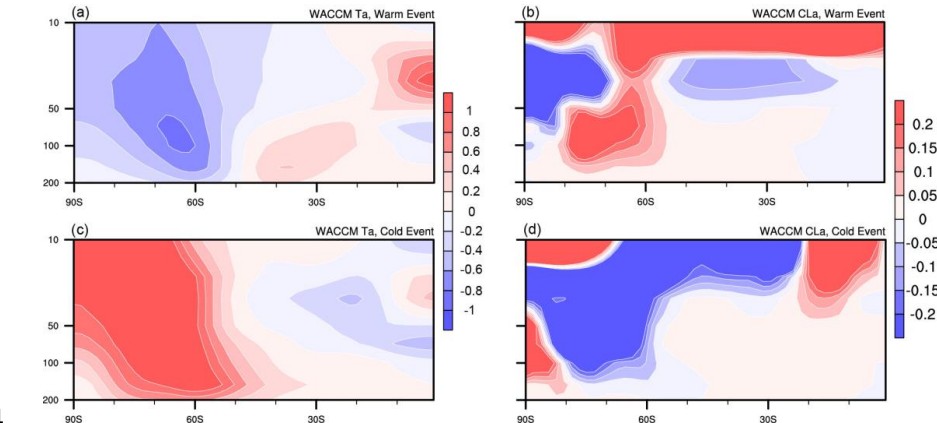

2  **Figure 10.** Zonal mean difference in temperature (K) between (a) S2 and S1, and (c) S3 and S1. (b)

3  and (d), as (a) and (c), but for active chlorine (ppbv).



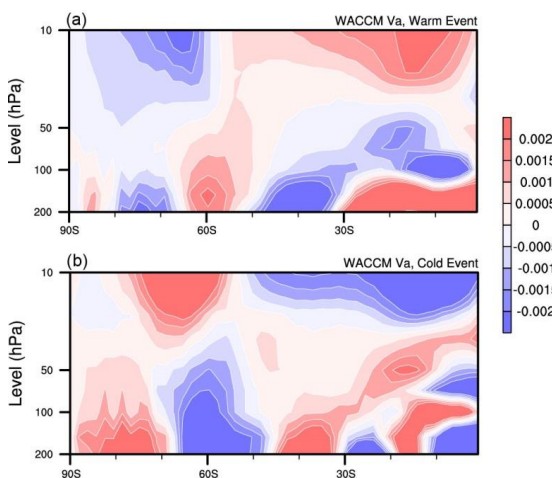

3    **Figure 11.** Zonal mean difference in meridional wind (m s$^{-1}$) between (a) S2 and S1, and (b) S3

4    and S1.





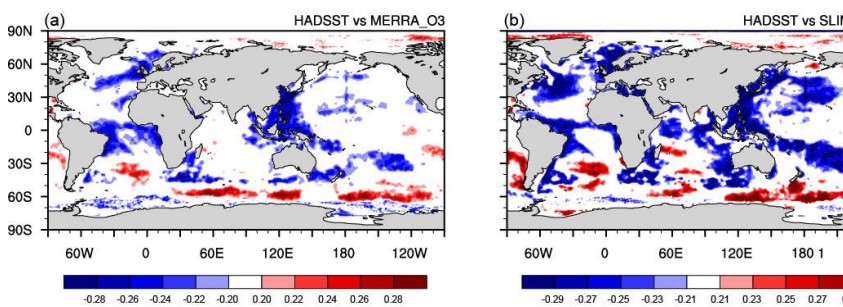

2 **Figure 12.** As Fig. 2, but with only the seasonal cycle removed before calculating the correlation

3 coefficients.





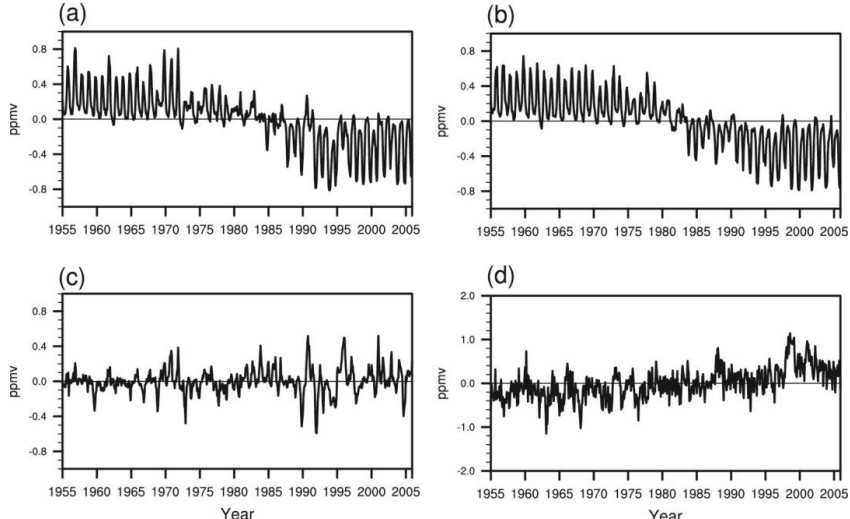

**Figure 13.** The southern high latitude lower stratospheric ozone variations averaged over the
region 60–90 °S at 200–50 hPa from T1 (a) and (T2+T3)/2 (b). (c) The difference in southern high
latitude lower stratospheric ozone variations between T1 and (T2+T3)/2. (d) SST variations (×–1)
in the marginal seas of East Asia (5 °S–35 °N, 100 °E–140 °E) based on the HadISST data. All
values are removed the seasonal cycle.