# Peer review of "The Relationship between Lower Stratospheric Ozone in the Southern High Latitudes and Sea Surface Temperature in the East Asian Marginal Seas in Austral Spring"

_Atmospheric Chemistry and Physics, 2016_

## Referee Comment (RC1) · Anonymous Referee #1 · 13 Feb 2017

This paper explores a possible linkage between the SST over East Asia Marginal Seas and the lower stratospheric ozone concentration over the Antarctica. The authors illustrated the correlation between the two from Reanalysis data and observation-constrained model simulations. They explained it by the SST-excite planetary waves that propagate across the equator and influence the stratosphere, and validated the mechanism by simulations of CESM. What the authors proposed is a novel mechanism for the coupling between the stratosphere and the surface. However, I have some concerns about the robustness of their results, and hope the authors can address them.

[Figure]

Main comments:

1.The ozone data. The ozone data employed in this study is from MERRA2 reanalysis and the TOMCAT/SLIMCAT simulations. While both data are constrained by observations in some way, they are not observations. Especially over southern high latitudes where direct observations are relatively sparse, the observational constrains would be quite weak. Since the study is build on the weak-but-significant correlation between the stratospheric ozone and the SST, it is important to establish the existence of such correlation and make sure it is not due to some artifact such as model biases.

Satellite observations of ozone is widely used in previous studies, which the authors claimed to have used in the abstract but is not even mentioned in the main text. I understand that most satellite observations of ozone has no coverage during polar night, but the largest variation of stratospheric ozone over Antarctica usually occurs in austral spring when photo-chemistry is active and the dynamical coupling with the troposphere is strong, and the satellite does cover this season. Another possibility is to use the ozone-sonde observations. Antarctic stations such as South Pole and Syowa maintain ozone-sonde observations back to 1960s. If the authors can show consistent results from these more direct observations, even if the correlations may not be as strong, it would greatly improve the confidence level for the proposed ozone-SST relation.

2. The authors showed results from time-slice and transient simulations and the quantify the contribution to ozone trends from SST warming based on these simulations. It is comforting to see these model simulations to be consistent with reanalysis, but the authors have not taken the full advantage of model simulations to establish the robustness of the results. Given that the SST-related stratospheric signal is usually small compared to the random internal variations, one may wonder if some of the signal appears just by chance. For example, the time series in Fig. 13(c) probably does not have a statistically significant trend judging by eyes. For the time-slice simulations, it should be straightforward to quantify the statistical significance. For the transient simulations, since the authors have two ensemble member (T2 and T3), it would be nice to show how different are the two member. If the difference between these two member is larger than the signal (T1-(T2+T3)/2), then the T1-(T2+T3)/2 may contain considerable contribution from random noise besides the SST over East Asia Marginal Seas, then the 17% contribution to ozone depletion trends may also be questionable.

Minor points:

It is not clear whether you are referring to the boreal or austral seasons throughout the paper since both hemispheres are involved in the paper. It is important to clarify this, because if the authors mean austral seasons, then the mechanisms do not work. The cross-equator propagation of anomalous planetary waves only occurs in austral summer and autumn, but that is the season when Southern Hemisphere stratosphere is dominated by easterlies and the prohibits vertical wave propagation into the stratosphere.

P6 L23: "Rieder et al. 2014, Zhang et al. 2015b" The authors cited these references to support the argument that MERRA2 ozone compares well with satellite observations. However, Rieder et al. did not even mention satellite observations. Zhang et al only compared data over China, which is not the region of interest in this paper.

P7 L7-9 "Figure 1 shows . . . past five decades" What do you mean by "ozone variations" here? Do you mean anomalies (with seasonal cycle removed)? Also, there is only 36 years not five decades.

P7 L11-13: The authors claimed the difference between MERRA2 and SLIMCAT ozone is small, but from the Fig. 1, the magnitudes of the difference is comparable to the magnitudes of ozone anomalies itself. Also, as the author stated later, "the regions of significant correlation are generally different for the two ozone datasets", which is another proof that the difference between the two datasets is not small.

P11 L12-14: "In addition, .. this location" From the Fig. 5, it looks like most wave rays

reaching 60S do not stop there, but refract to lower latitudes.

P13 L18: "zonal circulation" I think you mean meridional circulation or zonal-mean circulation. Note that the meridional circulation in the stratosphere is better described by the Transformed Eulerian Mean (TEM) velocity rather than the conventional meridional wind, because the TEM formula includes the contribution from eddies, which is often important in the stratosphere.

P14 L4-7: "This may be . . . wave activity." This discussion on how SST warming over East Asia Marginal Seas leads to a weaker wave activity in the Southern Hemisphere stratosphere is very puzzling. Can you support your arguments with more evidence, such as observations of convective activity?

Figure 9. Can you also show the EP flux anomalies in the troposphere? Does the tropospheric EP flux anomalies support your proposed mechanism?

Figure 13 caption: "SST variations (x-1)" I don't think you timed SST variations by -1 in the figure, as it shows increasing and warming trends. Also the unit for panel (d) should be K.
* * *

---

## Referee Comment (RC2) · Anonymous Referee #2 · 17 Feb 2017

**General**

In this study, the authors observed a correlation between sea surface temperature over East Asian marginal seas and ozone in the southern high latitude lower stratosphere (the targeted region) from assimilated MERRA data and a model simulation (SLIM-CAT). Using the WACCM model (WACCM4) and a defined temperature index from the HadISST data, they separated warm and cold events over the East Asian marginal seas and found distinct differences between the two groups of the events in ozone concentrations in targeted region. The model simulation further reveals large differences,

generally in the opposite directions, between the warm and clod events in dynamic and chemical conditions that modulate transport and formation/depletion of stratospheric ozone. Finally, the impact of such a connection on the ozone trend in the targeted region was quantified with a series of numerical experiments using WACCM4. The authors attributed 17% of decreasing ozone trend in the targeted region to increasing SST over the marginal seas of East Asia.

The authors proposed a hypothesis that establishes the connection between SST variation over the marginal seas of East Asia and ozone in the targeted region (P12-13). The paper is well written in articulating the connection and explaining the hypothesis. The proposed connection is novel and important. I suggest that the authors use some additional datasets to confirm or adjust their proposal. These data include ozonesonde data (there are about 7 stations over the Antarctic region during different periods), the TOST data (the Trajectory-mapped Ozonesonde dataset for the Stratosphere and Troposphere), and satellite data (although satellite data quality usually decreases with latitude).

For the long-term ozone trend over the targeted region, the ozone concentrations from MERRA, SLIMCAT should be compared with the WACCM simulation. More importantly, all simulated or assimilated data can be compared with observations so the estimated 17% contribution of increasing SST over the marginal seas of East Asia to the ozone trend in the targeted region can be confirmed or refined.

The authors also looked into the seasonal variation of the proposed connection in some aspects (P13, L 20-22, Figures 5-6). Does removing the seasonal cycle enhance or smooth the signal of this connection? As some lags appear in the MERRA data (Figure 4), will the connection be more significant if the authors use monthly data with consideration of the lags?

Specific

P2, L2, no satellite data are directly used in this study.

P3, L2, in this and other places (e.g., P7, L8-9, P16, L7-9), the authors stated similar sentences like "Ozone variations over recent decades exhibit a strong decreasing trend...". This may be the case for the Antarctic region, which is the focus of this study. It is not necessarily the case for other regions. In the Northern Hemisphere, stratospheric ozone recovery has been observed since the late 1990s after the Montreal Protocol and its amendments, although some surprising declines in ozone there were observed in recent years. So, please be specific.

P6, L4, what is the horizontal resolution for MERRA2 data used in this study, $2°\times2.5°$? How about SLIMCAT?

P8, L4-6, what is the significant level? 90%? 95%?

P8, L17, add "phi (using the Greek letter) is latitude".

P13, L20-22, not shown?

P14, L4-6, please rephrase the sentence.

P11, L11, use "further support" to replace "validate".

P18, L19, "observation"? No observation data are directly shown in this paper. The MERRA data may not be taken as "observation".

Indicate whether boreal or austral seasons (including months) are referred earlier in the paper.

Figure 1, is the ozone variation the same as or different from ozone anomaly? Is this normalized ozone anomaly? Are the seasonal cycle and trend removed? If not, add a trend to the figure. Indicate if the trend is significant. The variation is not straightly downward. A slight increase in ozone appears during 2010-2015.

Figures 2, 6, and 10, what is the significant level?

Figure 4, the label for the x-axis is Lag (month). Is the long-term trend removed from

the data?

Figure 6, indicate these are boreal or austral spring, summer, autumn and winter, including months. Should the seasonal cycles be removed?

Figures 7-11, the y-axis is not in the same format. Some have no unit, and some no label.

Figure 9, the annotation for the arrow is too small to see clearly.

Figure 13, the unit for SST variation should be K as shown in an earlier version. Also, please provide label for the y-axes. Is the ozone variation the same as Figure 1? Or the ozone trend in Figure 1 is removed? Why are they different?

There are a few inconsistency in the reference format. For example, some capitalize each word, some not.

---

## Author Comment (AC1) · 24 Mar 2017

Please see the supplement.

Please also note the supplement to this comment:
http://www.atmos-chem-phys-discuss.net/acp-2016-1053/acp-2016-1053-AC1-supplement.zip

---

## Author Comment (AC2) · 24 Mar 2017

Please see the supplement.

Please also note the supplement to this comment:
http://www.atmos-chem-phys-discuss.net/acp-2016-1053/acp-2016-1053-AC2-supplement.zip

---

## Author Response (AR1)

**Responses to Referees**

**Manuscript number**: acp-2016-1053

**Title:** The Relationship between Lower Stratospheric Ozone in the Southern High Latitude and Sea Surface Temperature in the East Asia Marginal Seas

**Author(s)**: Wenshou Tian, Yuanpu Li, Fei Xie, Jiankai Zhang, Martyn P. Chipperfield, Wuhu Feng, Sen Zhao, Xin Zhou, Yun Yang, Xuan Ma

**Summary of changes for Referees and the Editor**

We sincerely thank the reviewers for their important comments and the editor for their kind assistance with our manuscript. The major revisions are summarized as follows:

1. In the revised paper, we focus on investigating the relationship between lower stratospheric ozone in the southern high latitudes and sea surface temperature in the East Asian Marginal Seas **only** in austral spring. The title of the manuscript has been changed to "The Relationship between Lower Stratospheric Ozone in the Southern High Latitudes and Sea Surface Temperature in the East Asian Marginal Seas in Austral Spring".

2. Using two kinds of satellite ozone data (SWOOSH and GOZCARDS) to investigate the relationship between lower stratospheric ozone in the southern high latitudes and sea surface temperature in the East Asian Marginal Seas. The results are in good agreement with those from the MERRA2 reanalysis and SLIMCAT output. This improves the confidence level of the ozone–SST relationship in our study.

3. The significance of the results from the WACCM4 outputs is tested.

**Response to Referee 1**

*This paper explores a possible linkage between the SST over East Asia Marginal Seas and the lower stratospheric ozone concentration over the Antarctica. The authors illustrated the correlation between the two from Reanalysis data and observation constrained model simulations. They explained it by the SST-excite planetary waves that propagate across the equator and influence the stratosphere, and validated the mechanism by simulations of CESM. What the authors proposed is a novel mechanism for the coupling between the stratosphere and the surface. However, I have some concerns about the robustness of their results, and hope the authors can address them.*

**Response:** Thanks to the reviewer for sparing time to go through the manuscript, highlighting very important issues and providing helpful comments and valuable suggestions to improve the manuscript. We have revised the manuscript carefully according to the reviewer's comments and suggestions. The detailed point-to-point responses to the reviewer's comments are listed as follows.

*Main comments:*

**1.** *The ozone data. The ozone data employed in this study is from MERRA2 reanalysis and the TOMCAT/SLIMCAT simulations. While both data are constrained by observations in some way, they are not observations. Especially over southern high latitudes where direct observations are relatively sparse, the observational constrains would be quite weak. Since the study is build on the weak-but-significant correlation between the stratospheric ozone and the SST, it is important to establish the existence of such correlation and make sure it is not due to some artifact such as model biases. Satellite observations of ozone is widely used in previous studies, which the authors claimed to have used in the abstract but is not even mentioned in the main text. I understand that most satellite observations of ozone has no coverage during polar night, but the largest variation of stratospheric ozone over Antarctica usually occurs in austral spring when photo-chemistry is active and the dynamical coupling with the troposphere is strong, and the satellite does cover this season. Another possibility is to use the ozone-sonde observations. Antarctic stations such as South Pole and Syowa maintain ozone-sonde observations back to 1960s. If the authors can show consistent results from these more direct observations, even if the correlations may not be as strong, it would greatly improve the confidence level for the proposed ozone-SST relation.*

**Response:** Thanks for this important comment. In the revised paper, we have added two types of satellite ozone data to investigate the relationship between lower stratospheric ozone in the southern high latitudes and sea surface temperature in the East Asian Marginal Seas. One is the Global OZone Chemistry And Related trace gas

Data records for the Stratosphere (GOZCARDS) (Froidevaux et al. 2015) and the other is the Stratospheric Water and OzOne Satellite Homogenized (SWOOSH) ozone satellite data (Davis et al. 2016). The zonal mean satellite-based GOZCARDS is produced from high quality data from past missions (e.g., SAGE, HALOE data) as well as ongoing missions (ACE-FTS and Aura MLS). Its meridional resolution is 10 ° with 25 pressure levels from the surface up to 0.1 hPa. The zonal mean SWOOSH dataset is a merged record of stratospheric ozone and water vapor measurements taken by a number of limb sounding and solar occultation satellites (SAGE-II/III, UARS HALOE, UARS MLS, and Aura MLS instruments). Its meridional resolution is 2.5 ° with 31 pressure levels from 300 to 1 hPa.

Figure R1 shows the correlation coefficients between southern high latitude lower stratospheric ozone variations from the four ozone datasets and SST from HadISST in austral spring. It is apparent that the regions of significant correlation are generally different for the four ozone datasets except for the East Asian Marginal Seas; i.e., 5 °S–35 °N, 100 °E–140 °E, where the most significant correlations between Antarctic stratospheric ozone variations and SST are seen in all four ozone datasets. This result improves the confidence level of the ozone–SST relationship in our study.

Figure R1 is Figure 2 in the revised paper.

[Figure]

**Figure R1**. Correlation coefficients between southern high latitude lower stratospheric ozone variations and SST from HadISST in austral spring. The ozone variations are averaged over the region 60–90 °S at 200–50 hPa in austral spring. (a) Ozone from MERRA2 and (b) ozone from SLIMCAT for 1979–2015. (c) Ozone from GOZCARDS for 1979–2012. (d) Ozone from SWOOSH for 1984–2015. Only regions with statistical significance above the 95% confidence level are colored; statistical significance is calculated using the two-tailed Student's *t*-test and the $N^{\text{eff}}$ of DOF. The seasonal cycles and linear trends were removed prior to calculating the correlation coefficients.

References:

Davis S M et al. 2016, The Stratospheric Water and Ozone Satellite Homogenized (SWOOSH) database: A long-term database for climate studies, *Earth Syst. Sci. Data*, **8**, 461–490.

Froidevaux L et al. 2015, Global OZone Chemistry And Related trace gas Data records for the Stratosphere (GOZCARDS): methodology and sample results with a focus on HCl, H2O, and O3, *Atmos. Chem. Phys.* **15(18)**, 10471–10507.

**2.** *The authors showed results from time-slice and transient simulations and the quantify the contribution to ozone trends from SST warming based on these simulations. It is comforting to see these model simulations to be consistent with reanalysis, but the authors have not taken the full advantage of model simulations to establish the robustness of the results. Given that the SST-related stratospheric signal is usually small compared to the random internal variations, one may wonder if some of the signal appears just by chance. For example, the time series in Fig. 13(c) probably does not have a statistically significant trend judging by eyes. For the time-slice simulations, it should be straightforward to quantify the statistical significance. For the transient simulations, since the authors have two ensemble member (T2 and T3), it would be nice to show how different are the two member. If the difference between these two member is larger than the signal (T1-(T2+T3)/2), then the T1-(T2+T3)/2 may contain considerable contribution from random noise besides the SST over East Asia Marginal Seas, then the 17% contribution to ozone depletion trends may also be questionable.*

**Response:** Thanks for this comment. In the revised paper, we have calculated the significance of the results from WACCM4 simulations.

For the trend of the time series in Fig. 13c, the corresponding Table 4 in the original manuscript has been revised as Table R1 below. Note that the significance of the trends has been calculated.

**Table R1**. Linear trends of ozone variations over the region 200–50 hPa and 60–90 °S from experiments with (T1) and without SST (T2 +T3) variations in the East Asian Marginal Seas (T1–3 see Table 3).

| Experiments | Values |
|---|---|
| Linear trend of ozone variations over the region 200–50 hPa and 60–90 °S from T1 (Trend1) | -1.2 $\times 10^{-3}$ ppmv/month[**] |
| Linear trend of ozone variations over the region 200–50 hPa and 60–90 °S from (T1 – (T2+T3)/2) (Trend2) | -0.204 $\times 10^{-3}$ ppmv/month[*] |

[**]: the trend is significant at 99% confident level. [*]: the trend is significant at 95% confident level. The calculation of the statistical significance of the trend uses the two-tailed Student's *t*-test.

[Figure]

**Figure R2**. The differences in southern high latitude lower stratospheric ozone variations between T2 and T3 (green) and between T1 and (T2+T3)/2) (black), averaged over the region 60–90°S at 200–50 hPa.

Figure R2 shows the differences in southern high latitude lower stratospheric ozone variations between T2 and T3 and between T1 and (T2+T3)/2). It is evident that the difference between T2 and T3 is much smaller than that between T1 and (T2+T3)/2). This means that the difference between T1 and (T2+T3)/2) is not random but mainly originates from the SST forcing over the East Asian Marginal Seas.

For the results from the time-slice simulations, the statistical significance of the simulated anomalies is also calculated using the two-tailed Student's *t*-test. Figures 8–11 in the original manuscript have been replotted as Figures R3–R6 below.

[Figure]

**Figure R3.** Zonal mean differences in ozone (ppmv) in austral spring between WACCM simulations (a) S2 and S1, and (b) S3 and S1. Statistical significance above 95% confident level is stippled. Statistical significance of the simulated anomalies is calculated using the two-tailed Student's t-test.

[Figure]

**Figure R4.** Differences in E–P flux vectors (black arrows) and divergence (color shading) in austral spring between (a) S2 and S1, and (c) S3 and S1. Units for the horizontal and vertical

vector directions are $10^7$ and $10^5$ kg s$^{-1}$, respectively. (b) and (d), as (a) and (c), but for zonal wind (m s$^{-1}$). Statistical significance above 95% confident level is stippled.

[Figure]

**Figure R5.** Zonal mean difference in temperature (K) in austral spring between (a) S2 and S1, and (c) S3 and S1. (b) and (d), as (a) and (c), but for active chlorine (ppbv). Statistical significance above 95% confident level is stippled.

[Figure]

**Figure R6.** Zonal mean difference in meridional wind (m s$^{-1}$) in austral spring between (a) S2 and S1, and (b) S3 and S1. Statistical significance above 95% confident level is stippled.

*Minor points:*

**1.** *It is not clear whether you are referring to the boreal or austral seasons throughout the paper since both hemispheres are involved in the paper. It is important to clarify this, because if the authors mean austral seasons, then the mechanisms do*

*not work. The cross-equator propagation of anomalous planetary waves only occurs in austral summer and autumn, but that is the season when Southern Hemisphere stratosphere is dominated by easterlies and the prohibits vertical wave propagation into the stratosphere.*

**Response:** Thanks for the comment. When we discuss the mechanism of cross-equatorial propagation of anomalous planetary waves, the seasons are boreal summer and autumn. In the revised paper, we have unified the description of the season in the whole manuscript. Sorry for the confusion.

**2.** *P6 L23: "Rieder et al. 2014, Zhang et al. 2015b" The authors cited these references to support the argument that MERRA2 ozone compares well with satellite observations. However, Rieder et al. did not even mention satellite observations. Zhang et al only compared data over China, which is not the region of interest in this paper.*

**Response:** Thanks for the comment. We have deleted these two references and added Wargan et al. (2017) in the revised paper. In addition, we have replotted Figure 1, which now includes the comparison of MERRA2 ozone with satellite ozone observations. Please see Figure R7a below. The time series of southern high latitude lower stratospheric ozone variations from MERRA2 is significantly correlated with that from GOZCARDS ozone ($r = 0.54$) and SWOOSH ozone ($r = 0.50$). In austral spring, these correlation coefficients increase to 0.72 and 0.77, respectively (Figure R7b). This illustrates that the variations of southern high latitude lower stratospheric ozone from MERRA2 compare well with those from satellite observations.

[Figure]

**Figure R7**. (a) Time series of southern high latitude lower stratospheric ozone variations averaged over the region 60–90 °S at 200–50 hPa from the MERRA2 (black line), SLIMCAT (blue line), GOZCARDS (red line) and SWOOSH (green line) ozone datasets. (b) Same as (a), but only for austral spring. The seasonal cycles and linear trends are removed.

**Response:** Thanks for this important comment. The discussion is indeed puzzling. Figure R8 shows the correlation between SST changes in the East Asian Marginal Seas and OLR (outgoing longwave radiation) in four seasons. It is found that the correlation coefficients are the largest over East Asian Marginal Seas only in austral spring. This corresponds to the relationship we found between SST changes in the East Asian Marginal Seas and southern high latitude lower stratospheric ozone being very strong in austral spring.

However, it should be pointed out that the negative correlation coefficient between SST changes and OLR over the East Asian Marginal Seas does not support our argument for SST warming (cooling) over the East Asian Marginal Seas creating less (more) convection due to weakened (enhanced) sea–land contrast along the coastline of East Asia in austral spring. We have deleted this paragraph in the revised paper.

It is found that there is enhanced E-P flux from lower latitudes to southern high

latitudes in the SST warming event over the East Asian Marginal Seas (Figure R9a). However, this increased EP flux does not propagate upward into the stratosphere but downward to lower levels, and *vice versa* for the SST cooling event. Fig. R9 explains why SST warming (cooling) over the East Asian Marginal Seas leads to a weaker (stronger) wave activity in the Southern Hemisphere stratosphere.

This explanation and Figures R8 and R9 have been added in the revised paper.

[Figure]

**Figure R8.** Correlation coefficients between the ST_MSEA index and OLR from NOAA in (a) austral spring, (b) austral summer, (c) austral autumn, and (d) austral winter between 1975 and 2013. Only significant correlations are colored. The seasonal cycles and linear trends were removed before calculating the correlation coefficients.

[Figure]

**Figure R9.** Differences in E–P flux vectors (black arrows) and divergence (color shading) at 1000 to 100 hPa between (a) S2 and S1, and (c) S3 and S1. Units for the horizontal and vertical vectors are $10^7$ and $10^5$ kg s$^{-1}$, respectively. Details of experiments S1–3 are given in Table 2 in the manuscript.

**8.** *Figure 9. Can you also show the EP flux anomalies in the troposphere? Does the tropospheric EP flux anomalies support your proposed mechanism?*

**Response:** Please see Figure R9 in above **Response 7**.

**9.** *Figure 13 caption: "SST variations (x-1)" I don't think you timed SST variations by -1 in the figure, as it shows increasing and warming trends. Also the unit for panel (d) should be K.*

**Response:** Thanks for the careful check. The Figure 13 has been replotted in the revised paper.

**Response to Referee 2**

*In this study, the authors observed a correlation between sea surface temperature over East Asian Marginal Seas and ozone in the southern high latitude lower stratosphere (the targeted region) from assimilated MERRA data and a model simulation (SLIMCAT). Using the WACCM model (WACCM4) and a defined temperature index from the HadISST data, they separated warm and cold events over the East Asian Marginal Seas and found distinct differences between the two groups of the events in ozone concentrations in targeted region. The model simulation further reveals large differences,generally in the opposite directions, between the warm and clod events in dynamic and chemical conditions that modulate transport and formation/depletion of stratospheric ozone. Finally, the impact of such a connection on the ozone trend in the targeted region was quantified with a series of numerical experiments using WACCM4. The authors attributed 17% of decreasing ozone trend in the targeted region to increasing SST over the marginal seas of East Asia. The authors proposed a hypothesis that establishes the connection between SST variation over the marginal seas of East Asia and ozone in the targeted region (P12-13). The paper is well written in articulating the connection and explaining the hypothesis. The proposed connection is novel and important. I suggest that the authors use some additional datasets to confirm or adjust their proposal. These data include ozonesonde data (there are about 7 stations over the Antarctic region during different periods), the TOST data (the Trajectory-mapped Ozonesonde dataset for the Stratosphere and Troposphere), and satellite data (although satellite data quality usually decreases with latitude). For the long-term ozone trend over the targeted region, the ozone concentrations from MERRA, SLIMCAT should be compared with the WACCM simulation. More importantly, all simulated or assimilated data can be compared with observations so the estimated 17% contribution of increasing SST over the marginal seas of East Asia to the ozone trend in the targeted region can be confirmed or refined.*

**Response:** We thank the reviewer for the positive evaluation of our study and we sincerely appreciate the reviewer's very helpful comments, which have helped us to greatly improve our paper. We have revised the manuscript carefully according to the reviewer's comments and suggestions.

According to the referee's comment, the major revision is using the observed ozone to confirm the relationship between SST changes in the East Asian Marginal Seas and southern high latitude lower stratospheric ozone. Since the satellite ozone observations cover wider range compared with Ozonesonde data, we have added two types of satellite ozone data to investigate the relationship between the lower stratospheric ozone in the southern high latitudes and the sea surface temperature in the East Asian Marginal Seas. One is the Global OZone Chemistry And Related trace gas Data records for the Stratosphere (GOZCARDS) (Froidevaux et al. 2015) and the

other is the Stratospheric Water and OzOne Satellite Homogenized (SWOOSH) ozone satellite data (Davis et al. 2016). The zonal mean satellite-based GOZCARDS is produced from high quality data from past missions (e.g., SAGE, HALOE data) as well as ongoing missions (ACE-FTS and Aura MLS). Its meridional resolution is 10° with 25 pressure levels from the surface up to 0.1 hPa. The zonal mean SWOOSH dataset is a merged record of stratospheric ozone and water vapor measurements taken by a number of limb sounding and solar occultation satellites (SAGE-II/III, UARS HALOE, UARS MLS, and Aura MLS instruments). Its meridional resolution is 2.5° with 31 pressure levels from 300 to 1 hPa.

Figure RR1 shows the correlation coefficients between southern high latitude lower stratospheric ozone variations from the four ozone datasets and SST from HadISST in austral spring. It is apparent from Figure RR1 that the regions of significant correlation are generally different for the four ozone datasets except for the East Asian Marginal Seas; i.e., 5°S–35°N, 100°E–140°E, where the most significant correlations between Antarctic stratospheric ozone variations and SST are seen in all four ozone datasets. This result improves the confidence level of the ozone–SST relationship in our study.

Figure RR1 is Figure 2 in the revised paper.

[Figure]

**Figure RR1**. Correlation coefficients between southern high latitude lower stratospheric ozone variations and SST from HadISST in austral spring. The ozone variations are averaged over the region 60–90°S at 200–50 hPa in austral spring. (a) Ozone from MERRA2 and (b) ozone from SLIMCAT for 1979–2015. (c) Ozone from GOZCARDS for 1979–2012. (d) Ozone from SWOOSH for 1984–2015. Only regions with statistical significance above the 95% confidence level are colored; statistical significance is calculated using the two-tailed Student's $t$-test and the $N^{eff}$ of DOF. The seasonal cycles and linear trends were removed prior to calculating the correlation coefficients.

**Response**: The reference format are checked and corrected. Thanks.

[revised manuscript text omitted]

---

## Referee Report (RR1)

"The Relationship between Lower Stratospheric Ozone in the Southern High Latitudes and Sea Surface Temperature in the East Asian Marginal Seas in Austral Spring" by Tian et al.

This revision has addressed the questions in my earlier review.

There are some minor issues.

1. Figure 3, the color bar is missing.
2. Figure 1, label ozone molecules in the y-axis using subscripts.
3. The authors answered a question related to the original Figure 4 but the figure was removed in this version, why?
4. Figure 1 shows a time series of ozone variations, removing seasonal cycles and linear trends in MERRA2, SLIMCAT, 3 GOZCARDS and SWOOSH. May the original and absolute ozone concentrations from these data sets be shown, as also asked by the other reviewer ?
5. The satellite datasets used in this study are based on SAGE, HALOE, ACE-FTS, and Aura MLS. Some of the data have no or very limited coverage in the southern polar region so the regional mean may not be extended to 90 ˚S. Please comment.

---

## Author Response (AR3)

**Responses to Editor and Referees**

**Manuscript number**: acp-2016-1053

**Title:** The Relationship between Lower Stratospheric Ozone in the Southern High Latitudes and Sea Surface Temperature in the East Asian Marginal Seas in Austral Spring

**Author(s)**: Wenshou Tian, Yuanpu Li, Fei Xie, Jiankai Zhang, Martyn P. Chipperfield, Wuhu Feng, Yongyun Hu, Sen Zhao, Xin Zhou, Yun Yang, Xuan Ma

**Response to Editor**

We sincerely appreciate the editor's very helpful comments and we are so sorry for not well considering the two questions pointed out by editor. Our detailed replies are listed below:

*P10 L13-14: The reviewer points out to be careful with the term "causality" Indeed, a correlation can never confirm or imply causality. Even if a model result suggests "causality" I suggest formulating more carefully here and avoid "confirm or imply". The model could give an indication.*

**Response:** We fully agree with this suggestion of the editor.

In the **Reponses file,** we have reformulated the relevant sentences following editor's comment (please see below the second Response in **Response to Referee 1**).

In the **manuscript**, we have deleted the term "causality" as well as the associated word "confirm or imply". In addition, we also weakened the expression of some sentences in the manuscript. For example: P2 L15-18: "The model simulations also reveal that approximately 17% of the decreasing trend in the southern high latitude lower stratospheric ozone observed over the past five decades **can be attributed to** the increasing trend in SST over the East Asian Marginal Seas." The "**can be attributed to**" has been changed to **"may be associated with".** Other similar sentences in the manuscript are also revised as this way.

*A review comment pointed out: "The satellite datasets used in this study are based on SAGE, HALOE, ACE-FTS, and Aura MLS. Some of the data have no or very limited coverage in the southern polar region so the regional mean may not be extended to 90 °S." As a response you calculate the correlation coefficients of ozone variations between averaged over 60–90 °S at 200–50 hPa and over 60–75 °S at 200–50 hPa from four datasets. You find that all these correlation coefficients are very large and conclude that the results are not sensitive to the average latitude range.*

*I think that you find a high correlation because the data sets 60–90 °S and 60–75 °S are identical to a large extent. Indeed the results are not very sensitive to the average latitude range. But in my view this would argue for using the 60–75 °S range*

*rather than extending the data to 90° where there are no measurements. Don't you just "dilute" the actual information from measurements with fill values up to 90°?*

**Response:** Thanks for the important comment. In the revised manuscript, we have performed the average between the 60–75°S latitude range instead of the 60–90°S latitude range in calculations of two satellite ozone data. The relevant Figures 1 and 2 have been replotted. The results from the new Figs. 1 and 2 are similar with original results.

Note that the corresponding Response in the **Responses to Referee 2** is also revised.

**Response to Referee 1**

*This is the second review of this manuscript. The authors have made significant improvements. I especially appreciated their efforts to include satellite observations into the analysis. This addition indeed strengthen the paper. I have a few minor comments for the authors to consider.*

**Response:** We thank again the reviewer for the helpful comments and valuable suggestions to improve the manuscript. We have revised the manuscript according to the reviewer's comments and suggestions. The detailed point-to-point responses to the reviewers' comments are listed as follows.

*P10 L10-11: please consider reword this sentence.*

*P10 L13-14: "it is first necessary to confirm the causality of this connection". I don't see anything in the following text that can confirm the causality.*

**Response:** Thanks for the comments. In our first version of the manuscript, there is a lead-lag correlation figure to preliminary understand the possible causality of connection between **monthly** SST and Antarctic stratospheric ozone variations. In our second version, we only focus on investigating the connection in austral spring (**three-month average**). There is no better way to calculate the lead-lag correlation. Thus, we deleted the lead-lag correlation figure in the second version. The lines 10-14 in page 10 in the manuscript are also redundant, which are deleted in the revised manuscript. Note that the model simulations presented in the section 4 of the manuscript maybe give an indication to understand the connection between SST and Antarctic stratospheric ozone.

*P12 L3-5: I don't quite follow the logic here. Why wave rays refracting to lower latitudes implies upward propagation?*

**Response:** Thanks for the comment. The sentence is a bit confusing. We have deleted the sentence "implying that the pathway of upward propagation of tropospheric waves from the marginal seas of East Asia possibly extends to 60 °S".

*Fig. 5: Can you compare the teleconnection pattern with the climatology, especially the lower wavenumber components? The interference of the teleconnection pattern and the climatological eddies might explain the connection between the SST over East Asian Marginal Seas and the wave anomalies in the Southern Hemisphere, which remains unclear in the current manuscript. Similar mechanism is used to explained the connection between ENSO and northern polar vortex by Garfinkel and Hartmann (2008).*

**Response:** Thanks for the important comment. This comment well helps us explain the connection between the SST over East Asian Marginal Seas and the wave anomalies in the Southern Hemisphere.

Following the Garfinkel and Hartmann (2008), Figure R1 shows the replotted Figure 5 in the manuscript, in which the climatological wave 1 in each season is overploted. It is apparent that positive/negative correlation coefficients correspond to positive/negative climatological wave 1 phases over Indo-Pacific warm pool but negative/positive climatological wave 1 phases in the middle and high latitudes of Southern Hemisphere in austral spring (Fig. R1a). The results in Fig. R1 implies that warm/cold SST events over East Asian Marginal Seas would increase/decrease the planetary wave activity at lower latitudes but decrease/increase the planetary wave activity at middle and high latitudes of Southern Hemisphere. The replotted Fig. 5 is in well agreement with the results in the study.

Above points have added and the reference has been cited in the revised paper.

**Reference:**

Garfinkel, C. I., and D. L. Hartmann, 2008: Different ENSO teleconnections and their effects on the stratospheric polar vortex, JGR, 113, D18114, DOI: 10.1029/2008JD009920.

[Figure]

**Figure R1.** Correlation coefficients between the ST_MSEAI and 300-hPa geopotential height (contour level) associated with stationary waves of wavenumber 1 (color) from the ERA-Interim reanalysis in (a) austral spring, (b) austral summer, (c) austral autumn, and (d) austral winter between 1979 and 2015. Only statistical significance above 95% confidence level is colored. The seasonal cycles and linear trends were removed before calculating the correlation coefficients.

*P12 L21-24: I suggested looking at OLR last time because the authors suggested the anomalous waves are generated from convection, and OLR is an index for convection. I don't think OLR can represent "generated wave activity". It is not clear to me what Fig. 6 intend to show.*

**Response:** Thanks for the comment. The term is a bit misleading. The OLR can represent convective activity in the lower latitudes, while stronger convective activity often corresponds to enhanced wave activity. The text is rephrased in the revised text.

*P13 L21-22: I don't see how Figs. 4 and 5 show that warm SST depress planetary wave activity.*

**Response:** Thanks for the comment. Figure 5 is replotted in the revised paper following the above comment, which implies that warm SST anomalies depress planetary wave activity now.

*P14 L17: It may be clearer to point out that the climatological v\* is negative (southward) in the Southern Hemisphere, so that a positive correlation with SST indicates a weakening of the stratopheric circulation associated with SST warming. It is also not clear whether the authors are referring to the negative correlations poleward of 60S or the positive correlation centered at 30S.*

**Response:** Thanks for the comments. The first point of the reviewer has been added in the revised manuscript following referee's suggestion. For the second comment, the relevant text is rephrased as "are positively correlated with lower stratospheric TEM v\* between 30 ℃S and 60 ℃S".

*P16 L2: "stronger convergence of the EP flux" It should be weaker wave breaking (convergence of EP flux).*

**Response:** Thanks for your careful check. It should be "stronger divergence of the EP flux". Corrected.

*P16 L19-21: It seems from Fig. 11 that correlation with v\* is opposite below and above 20 hPa, while polar vortex does not show such dependence on height. It also looks quite different from the reanalysis (Fig. 7d). Can you comment on that? Also, since ozone concentration is higher above 20 hPa, I expect that the horizontal transport by v\* above 20 hPa is more important than that below 20 hPa. Then we see more southward v\* across 60S during the warm events, bringing more ozone from mid-latitues into the polar region, which is opposite to what is stated here.*

**Response:** Thanks for the comment. Figure R2 combines the Figure 7d and Figure 11a of the manuscript. See the red color between 60 ℃S and 0 and blue color between 60 ℃S and 90 ℃S in Fig. 7d (Fig. R2a) and Fig. 11a (Fig. R2b), the patterns in the two figures are actually very similar. It needs to point out that the positive center below 50 hPa near 60 ℃S in Fig. 11a (Fig. R2b) is more southward than that in Fig. 7d (Fig. R2a). It makes the correlations below and above 20 hPa in Fig. 11a (Fig. R2b) seem to be opposite. The difference between Fig. R2a and R2b may be because the simulated polar vortex isn't completely same with observations.

In Fig. 11a (Fig. R2b), there are different responses of v\* below and above 20

hPa at 60 °S to SST anomalies over the East Asian Marginal Seas in simulations. It may be because the changes of v* below 20 hPa mainly caused by SST-induced polar vortex anomalies while v* variations above 20 hPa result from SST-induced BD circulation anomalies.

At lines 19–21of page 16, we mainly discuss the dynamical transport blow 50 hPa where is the region this study focusing on. We have clarified this information in the revised paper. Stronger southerly v* at 60 °S above 20 hPa during the SST warm events (Fig. R2b) would bring more ozone from mid-latitudes into the South Pole at upper stratosphere. See the Figure R3 (it is Figure 7a in the revised manuscript), the ozone above 20 hPa at South Pole is indeed increased during the SST warm events which is associated with the Fig. 11a (Fig. R2b).

[Figure]

**Figure R2.** A combination of Figure 7d (a) and Figure 11a (b) of the manuscript. (a) Correlation coefficients between ST_MSEAI and zonal mean TEM v* in austral spring (the southward climatological TEM v* is negative). The seasonal cycles and linear trends were removed before calculating the correlation coefficients. (b) Zonal mean TEM meridional wind (m s$^{-1}$) anomalies caused by SST warm events in the East Asian Marginal Seas in austral spring from simulations. Only statistical significance above 95% confidence level is colored.

[Figure]

**Figure R3**. Zonal mean differences in ozone (ppmv) in austral spring during the SST warm events by WACCM simulations

*P38 Table4: Can you also list trends from T2 and T3, separately? This will give readers a sense how large the internal variability may be and how variable the 17% is.*

**Reponse:** Thanks for the comment. The two trends have been added in Table 4 as below:

**Table 4.** Linear trends of ozone variations over the region 200–50 hPa and 60–90 °S from experiments with (T1) and without SST (T2 +T3) variations in the East Asian Marginal Seas (T1–3 see Table 3).

| Experiments | Values |
|---|---|
| Linear trend of ozone variations over the region 200–50 hPa and 60–90 °S from T1 (Trend1) | -1.2 $\times 10^{-3}$ ppmv/month[**] |
| Same as Trend1, but from T2 (Trend 2) | -1.0 $\times 10^{-3}$ ppmv/month[*] |
| Same as Trend1, but from T3 (Trend 3) | -0.89 $\times 10^{-3}$ ppmv/month[*] |
| Same as Trend1, but from (T1 − (T2+T3)/2) (Trend1_23) | -0.2$\times 10^{-3}$ ppmv/month[*] |

[**]: the trend is significant at 99% confidence level. [*]: the trend is significant at 95% confidence level. The calculation of the statistical significance of the trend uses the two-tailed Student's *t*-test.

*P45 L3: "meridional wind" I thought you are using TEM v\*.*

**Response:** Revised. Thanks.

*P52 Fig. 14: Change the label to "-SST" or reverse the labels on the left y-axis so that negative values are at the top and positive values are at the bottom.*

**Response:** Figure 14 is replotted as below Figure R4 following the comment. Thanks.

[Figure]

**Figure R4.** The difference in southern high latitude lower stratospheric ozone variations between T1 and (T2+T3)/2) (black line) and SST variations in the marginal seas of East Asia (5 °S–35 °N, 100 °E–140 °E) based on the HadISST data (red line). The seasonal cycle is removed from two time series.

**Response to Referee 2**

*This revision has addressed the questions in my earlier review. There are some minor issues.*

**Response:** We thank the reviewer again for the very helpful comments, which have helped us to greatly improve our paper. We have revised the manuscript carefully according to the reviewer's comments and suggestions.

*1. Figure 3, the color bar is missing.*

**Response:** Added, thanks.

*2. Figure 1, label ozone molecules in the y-axis using subscripts.*

**Response:** Revised, thanks.

*3. The authors answered a question related to the original Figure 4 but the figure was removed in this version, why?*

**Response:** Thanks for the comments. In our first version of the manuscript, there is a lead-lag correlation figure to confirm the causality of connection between **monthly** Antarctic stratospheric ozone and SST variations. In our second version, we only focus on investigating the connection in austral spring (**three-month average**). There is no better way to calculate the lead-lag correlation. However, the causality of the connection can be indirectly confirmed by the model simulations presented in the section 4 of the manuscript. Thus, we deleted the lead-lag correlation figure in the second version. The lines 10-14 in page 10 in the manuscript are also redundant, which are deleted in the revised manuscript.

*4. Figure 1 shows a time series of ozone variations, removing seasonal cycles and linear trends in MERRA2, SLIMCAT, GOZCARDS and SWOOSH. May the original and absolute ozone concentrations from these data sets be shown, as also asked by*

*the other reviewer?*

**Response:** Thanks for the comment. Since this study only focuses on austral spring, the Figure 1a is replaced by the variations of original ozone concentrations in austral spring from these four datasets (See Figure RR1 below). We can see the original ozone concentrations from MEERA2 and SLIMCAT are somewhat lower than that from the GOZCARDS and SWOOSH (Fig. RR1a); however, the variabilities of ozone concentrations from these four datasets are similar (Fig. RR1b).

[Figure]

**Figure RR1.** (a) Time series of original and absolute ozone concentrations at southern high latitude lower stratosphere averaged over the region 60–90 °S at 200–50 hPa in austral spring from the MERRA2 (black line), SLIMCAT (blue line), GOZCARDS (red line) and SWOOSH (green line) ozone datasets. (b), Same as (a), but the ozone variations are removed the seasonal cycles and linear trends.

*5. The satellite datasets used in this study are based on SAGE, HALOE, ACE-FTS, and Aura MLS. Some of the data have no or very limited coverage in the southern polar region so the regional mean may not be extended to 90 °S. Please comment.*

**Response:** Thanks for the comment. In the revised manuscript, we have performed the average between the 60–75 °S range instead of the 60–90 °S range in calculation of satellite ozone. The relevant Figures 1 and 2 have been replotted. The results from the

new Figs. 1 and 2 are similar with original results.